# Semi-infinite Nonconvex Constrained Min-Max Optimization

**Cody Melcher**
School of Mathematical Sciences
The University of Arizona
Tucson, AZ 85721
cmelcher@arizona.edu

**Zeinab Alizadeh**
Systems and Industrial Engineering
The University of Arizona
Tucson, AZ 85721
zalizadeh@arizona.edu

**Lindsey Heitt**
Systems and Industrial Engineering
The University of Arizona
Tucson, AZ 85721
lhiett9@arizona.edu

**Afrooz Jalilzadeh**
Systems and Industrial Engineering
The University of Arizona
Tucson, AZ 85721
afrooz@arizona.edu

**Erfan Yazdandoost Hamedani**
Systems and Industrial Engineering
The University of Arizona
Tucson, AZ 85721
erfany@arizona.edu

## Abstract

Semi-Infinite Programming (SIP) has emerged as a powerful framework for modeling problems with infinite constraints, however, its theoretical development in the context of nonconvex and large-scale optimization remains limited. In this paper, we investigate a class of nonconvex min-max optimization problems with nonconvex infinite constraints, motivated by applications such as adversarial robustness and safety-constrained learning. We propose a novel inexact dynamic barrier primal-dual algorithm and establish its convergence properties. Specifically, under the assumption that the squared infeasibility residual function satisfies the Łojasiewicz inequality with exponent $\theta \in (0, 1)$, we prove that the proposed method achieves $\mathcal{O}(\epsilon^{-3})$, $\mathcal{O}(\epsilon^{-6\theta})$, and $\mathcal{O}(\epsilon^{-3\theta/(1-\theta)})$ iteration complexities to achieve an $\epsilon$-approximate stationarity, infeasibility, and complementarity slackness, respectively. Numerical experiments on robust multitask learning with task priority further illustrate the practical effectiveness of the algorithm.

## 1 Introduction

Recent advances in artificial intelligence (AI), particularly foundation models for language, vision, and multimodal reasoning, have revealed impressive capabilities and critical vulnerabilities at the same time. These models are often susceptible to adversarial perturbations, leading to concerns about their reliability and safety in high-stakes applications [26]. Similarly, ensuring robustness in domains such as supply chain management and autonomous control systems requires optimization frameworks that can account for worst-case scenarios across a continuum of uncertainties. Semi-Infinite Programming (SIP), which naturally models problems with infinite constraints, provides a powerful tool for addressing these challenges. However, despite its rich and extensive theoretical

39th Conference on Neural Information Processing Systems (NeurIPS 2025).

and algorithmic development, comparatively less attention has been devoted to designing efficient first-order methods for emerging applications in modern AI and Operations Research.

Motivated by this gap, in this paper, we consider the following min-max optimization with infinite constraints:

$$\min_{x \in \mathbb{R}^n} \max_{y \in Y} \phi(x, y), \quad \text{s.t.} \quad \psi(x, w) \leq 0, \quad \forall w \in W, \tag{1}$$

where $\phi : \mathbb{R}^n \times \mathbb{R}^m \to \mathbb{R}$ and $\psi : \mathbb{R}^n \times \mathbb{R}^\ell \to \mathbb{R}$ are continuously differentiable functions. Moreover, $Y \subseteq \mathbb{R}^m, W \subseteq \mathbb{R}^\ell$ are convex, non-empty, closed sets. Problem (1) arises in a wide range of applications, including robust optimization [5, 7], distributionally robust learning [59], and adversarial machine learning [50, 60]. In these settings, we aim to optimize against worst-case scenarios while ensuring that an infinite family of constraints is satisfied. Such constraints often encode safety, fairness, or robustness requirements that must hold uniformly over a continuous set of parameters.

To develop efficient algorithms for solving problem (1), it is critical to recognize the distinctive structural complexities it introduces and why existing approaches are insufficient. The problem combines features of min-max optimization and SIP, resulting in a constrained min-max optimization problem with an infinite set of nonlinear constraints. While min-max problems and SIPs have been studied independently, their intersection as in (1) poses unique challenges. Standard first-order algorithms for solving min-max problems typically assume finite-dimensional constraints without nonlinear constraints, whereas SIP approaches often assume convexity or lack the nested max structure in the objective. Moreover, the presence of inner maximizations in both the objective and constraints renders traditional gradient-based methods or constraint sampling techniques inadequate, particularly when $\phi$ and/or $\psi$ are nonconvex in $x$. This calls for new algorithmic strategies that can simultaneously handle the nonconvexity, the infinite constraint set, and the nested structure of the problem.

In the following, we review relevant literature in each of these areas to highlight existing methods and identify the challenges that arise in tackling problem (1).

## 1.1 Literature Review

**Nonconvex constrained optimization.** Consider the following constrained optimization problem

$$\min_{x \in X} f(x) \quad \text{s.t.} \quad g(x) \leq \mathbf{0}, \tag{2}$$

where $f : \mathbb{R}^n \to \mathbb{R}$ and $g : \mathbb{R}^n \to \mathbb{R}^m$ are continuously differentiable, but not necessarily convex, and $X \subseteq \mathbb{R}^n$ is a closed convex set. When $f(\cdot)$ is nonconvex and the constraints $g_i$'s are either linear or convex inequalities, a range of algorithms have been proposed, including penalty methods, Lagrangian methods, and augmented Lagrangian methods [30, 33, 28].The study of optimization algorithms for non-convex constrained problems has a long history, including analyses of the global asymptotic convergence of methods such as Augmented Lagrangian method [6], Augmented Lagrangian trust-region [14], and Sequential Quadratic Programming [20], among others. However, due to the nonconvexity of the constraints, these methods may converge to infeasible stationary points. To ensure convergence to feasible stationary points, additional assumptions are typically required. For instance, assuming access to a (nearly) feasible solution, several methods have been developed to find an $\epsilon$-KKT point within $\mathcal{O}(\epsilon^{-4})$ iterations [49, 40, 66]. With further regularity conditions, the complexity can be improved to $\mathcal{O}(\epsilon^{-3})$ for these methods and others, such as [39, 47].

More recently, the dynamic barrier gradient descent (DBGD) method [25] has emerged as a principled alternative, incorporating barrier functions that smoothly penalize the constraint violation. The proposed algorithm has been studied in continuous time and shown to achieve an $\mathcal{O}(1/t)$ convergence rate in terms of KKT residuals, assuming the dual iterates remain bounded. However, this assumption may not hold in practice, and when $\lambda_t$ is unbounded, the convergence slows down. In such cases, the KKT violation decays at a rate of $\mathcal{O}(\max(1/t^{2/\tau}, 1/t^{1-1/\tau}))$ where $\tau > 1$ is a user-defined parameter controlling the dynamic barrier.

**Min-max optimization.** Min-max optimization, rooted in von Neumann's foundational work [69], has become increasingly central in modern applications such as adversarial learning, fairness, and distributionally robust optimization [64, 22, 58]. Classical convex-concave problems have

been well-studied using primal-dual and gradient-based algorithms [18, 16, 27]. In recent years, nonconvex–strongly concave (NC-SC) problems have received significant attention, with algorithms such as gradient descent ascent (GDA) and alternating GDA achieving rates of $\mathcal{O}(\kappa^2 \varepsilon^{-2})$ [57, 73, 42], where $\kappa$ denotes the condition number. Proximal point methods combined with acceleration further improve the rate to $\tilde{\mathcal{O}}(\sqrt{\kappa}\varepsilon^{-2})$–a rate shown to be optimal under standard complexity assumptions [77, 38, 43]. For the more general nonconvex–concave (NC-C) setting, the convergence rates are typically reduced to $\mathcal{O}(\epsilon^{-4})$ due to the absence of strong concavity, e.g., see [48, 71, 11] and the references therein. Despite recent studies, existing methods assume convex and easy-to-project constraints. To the best of our knowledge, there is no work on nonconvex-(strongly) concave min-max problems with nonconvex nonlinear constraints.

**Robust Optimization.** Robust optimization (RO) provides a framework for decision making under uncertainty by introducing an uncertainty sets $Y, W$ and requiring any candidate solution $x$ remain feasible for all realizations in the uncertainty set, leading to problems of the form $\min_{x \in X} \sup_{y \in Y} f(x, y)$ s.t. $\sup_{w \in \mathcal{W}} g(x, w) \leq 0$. When suitable regularity conditions hold, the inner supremum admits a convex dual reformulation [5, 7]. In more general cases, RO is addressed through cutting-set methods [53, 4] or scenario-based approaches that approximate $Y, W$ with a finite sample [12, 13]. Building on RO, distributionally robust optimization (DRO) defines an ambiguity set $\mathcal{P}$ of probability distributions and yields the formulation $\min_x \sup_{P \in \mathcal{P}} \mathbb{E}_P[\ell(x, \xi)]$ [59, 36]. In structured convex scenarios, duality can transform the min–max problem to a single-level program [21, 52, 54]. However, these approaches do not extend to the general setting considered in this paper.

**Semi-Infinite Programming** Semi-infinite programming (SIP) was introduced in the 1960s through the foundational work of Charnes, Cooper, and Kortanek [17], and has since evolved into a versatile framework with applications in functional approximation [65], finance [19], and multi-objective learning [67]. For convex SIP, three main approaches have been developed: discretization [8, 61], cutting surface methods [29, 51], and penalty methods [41, 74]. These methods often entail solving non-trivial subproblems, which makes them computationally expensive in large-scale settings. On the theoretical side, duality and sufficient optimality conditions have been established [62, 34]. In contrast, the theory and algorithms for nonconvex SIP remain less developed, reflecting the greater difficulty of the nonconvex setting [23]. Recent work has explored new discretization strategies [68] and min–max reformulations of SIP constraints that motivate primal–dual algorithms [57, 31]. However, none of these approaches address our setting fully, with the closest being recent work by Yao et al. [75], which develops first-order primal–dual schemes with non-asymptotic guarantees under convex regime.

## 1.2 Applications

**Robust Multi-Task Learning with Task Priority.** Multi-task learning (MTL) is a paradigm in machine learning that aims to simultaneously learn multiple related tasks by leveraging shared information among them [15]. The key idea is to enhance generalization performance by enabling tasks to learn collaboratively rather than in isolation. Specifically, let $\{\mathcal{T}_i\}_{i=1}^T$ represent a collection of $T$ tasks, each associated with its own training dataset $\{\mathcal{D}_i^{tr}\}_{i=1}^T$, where the feature space is shared across tasks. Each task $\mathcal{T}_i$ is characterized by a loss function $\ell_i(x, \mathcal{D}_i^{tr})$, where $x$ denotes the shared parameters learned across all tasks. In applications where one task is prioritized, the problem can be formulated by minimizing its loss while enforcing the losses of the remaining tasks to stay below specified thresholds $r_i$. Most existing MTL formulations assume a uniform distribution over training samples when computing task-specific losses; however, in real-world applications, the underlying data distributions are often uncertain or unknown. To address this, one standard approach is to utilize the distributionally robust optimization (DRO) [59], and define the loss function for task $i$ as the weighted sum over the training dataset $\sum_{\xi_j \in \mathcal{D}_i^{tr}} y_j^{(i)} \ell_i(x, \xi_j)$ where the weights $\{y_j^{(i)}\}_{j=1}^{m_i}$ lies in an uncertainty set $Y_i$, e.g., $Y_i = \{y \in \Delta_{m_i} : V(y, \frac{1}{m_i}\mathbf{1}_{m_i}) \leq \rho\}$ where $V(Q, P)$ denotes the divergence measure between two sets of probability measures $Q$ and $P$ and $\Delta_m \triangleq \{y \in \mathbb{R}_+^m | \sum_{i=1}^n y_i = 1\}$ represents the simplex set [54]. This leads to a DRO-based MTL formulation with task prioritization:

$$\min_{x \in \mathbb{R}^m} \max_{y^{(1)} \in Y_1} \sum_{\xi_j \in \mathcal{D}_1^{tr}} y_j^{(1)} \ell_1(x, \xi_j)$$

$$\text{s.t.} \sum_{\xi_j \in \mathcal{D}_i^{tr}} y_j^{(i)} \ell_i(x, \xi_j) \leq r_i, \quad \forall y^{(i)} \in Y_i, \ \forall i \in \{2, \dots, T\}.$$

**Robust, Energy-Constrained Deep Neural Networks Training (DNN).** This problem involves optimizing model performance while limiting the total energy consumed during training [72]. This constraint arises in resource-constrained environments such as edge devices or large-scale systems where power efficiency is critical. In particular, the goal is to train a deep neural network that performs reliably under worst-case distribution uncertainty, while satisfying energy and sparsity constraints under hardware-induced uncertainty. Consider a neural network with $L$ layers and let $W = [w^{(1)}, \dots, w^{(L)}]$ denote the collection of weights of all the layers, and $S = [s^{(1)}, \dots, s^{(L)}]$ represent the collection of the non-sparse weights of all the layers. The goal is to train the model by minimizing worst-case loss under distribution uncertainty $y \in Y$, while ensuring compliance with resource constraints described by $\varphi_2(S) \le E_{\text{budget}}$. This problem can be formulated as follows:

$$\min_{(W,S)} \max_{y \in Y} \sum_{\xi_i \in \mathcal{D}_{tr}} y_i \ell(W; \xi_i) \quad \text{s.t.} \quad \varphi_1(w^{(u)}) \le s^{(u)}, \quad \varphi_2(S) \le E_{\text{budget}}, \quad \forall u \in \{1, \dots, L\}.$$

where $\varphi_1(w^{(u)})$ calculates the sparsity in layer with the sparsity level $s^{(u)}$, $\varphi_2(S)$ represents the energy consumption of the DNN, which is usually a non-convex function, and the constant $E_{\text{budget}}$ is the threshold of the maximum energy budget.

**Contributions.** In this paper, we study a class of semi-infinite constrained min-max optimization problems. Unlike existing methods, our framework accommodates both nonconvex objectives and constraints defined over infinite cardinality constraint sets, thereby significantly broadening the scope of both min-max optimization and semi-infinite programming (SIP). We propose a novel *Inexact Dynamic Barrier Primal-Dual* (iDB-PD) method, which performs gradient-based updates on the primal variables to simultaneously reduce the objective function and the infeasibility residual, formulated through a quadratic programming subproblem. To regulate the behavior of the search direction near the feasible region, we introduce an indicator function that ignores the constraint when it is satisfied and adjusts the direction to focus solely on minimizing the objective. Assuming that the squared inexact infeasibility residual $[\psi(\cdot, w)]_+^2$ satisfies the Łojasiewicz inequality with exponent $\theta \in (0, 1)$ for any $w \in W$, we establish the first global non-asymptotic convergence guarantees for the class of problems where the objective and constraint functions are smooth and are either strongly concave or satisfying Polyak-Łojasiewicz (PL) in their second component. In particular, we show that our method attains an $\epsilon$-KKT solution within $\mathcal{O}(\epsilon^{-3})$, $\mathcal{O}(\epsilon^{-6\theta})$, and $\mathcal{O}(\epsilon^{-3\theta/(1-\theta)})$ in terms of first-order stationarity, feasibility, and complementarity slackness, respectively. Finally, we demonstrate the effectiveness of our algorithm on real-world data by applying it to robust multi-task learning (MTL) with task priority across various datasets.

## 2 Preliminaries

This section introduces the notations, definitions, and assumptions used throughout the analysis.

**Notation.** Throughout the paper, $\|\cdot\|$ denotes the Euclidean norm. We use $\mathbb{R}_+^n$ to denote the nonnegative orthant. For $x \in \mathbb{R}^n$, we adopt $[x]_+ \in \mathbb{R}_+^n$ to denote $\max\{x, 0\}$, where the maximum is taken componentwise. For any convex set $C \subseteq \mathbb{R}^n$ and point $x \in C$, the normal cone is denoted by $\mathcal{N}_C(x) \triangleq \{p \in \mathbb{R}^n \mid p^T x \ge p^T y, \text{ for all } y \in C\}$. For a differentiable vector-valued function $f : \mathbb{R}^n \to \mathbb{R}^m$, the Jacobian is denoted by $\mathbf{J}f : \mathbb{R}^n \to \mathbb{R}^{m \times n}$ defined as the matrix of gradients.

Next, we define the Łojasiewicz property. Intuitively, this inequality controls the behavior of the gradient near critical points, preventing it from vanishing too quickly unless the function value is close to its critical value.

**Definition 2.1** (Łojasiewicz inequality). *Let $f : \mathbb{R}^n \to \mathbb{R}$ be a differentiable function on an open subset of $\mathbb{R}^n$, and let $x^* \in \mathbb{R}^n$ be a critical point of $f$, i.e., $\nabla f(x^*) = 0$. We say that $f$ satisfies the Łojasiewicz inequality at $x^*$ if there exist constants $\theta \in [0, 1)$, $c > 0$, and a neighborhood $U$ of $x^*$ such that for all $x \in U$,*

$$c |f(x) - f(x^*)|^\theta \le \|\nabla f(x)\|. \tag{3}$$

*The constant $\theta \in [0, 1)$ is called the Łojasiewicz exponent.*

Łojasiewicz property and its extension for nonsmooth functions [10] (also known as the Kurdyka-Łojasiewicz (KL) property) plays a central role in non-convex optimization and has been extensively

studied in the literature [1, 2, 70], including its special case for $\theta = 1/2$, also known as the Polyak-Łojasiewicz (PL) inequality [32]. The fundamental contributions to this concept are attributed to Kurdyka [37] and Łojasiewicz [45]. A broad class of functions has been shown to satisfy Łojasiewicz property, including real analytic functions [46], functions definable in $o$-minimal structures, and differentiable subanalytic functions [37]. In the context of DNN training, networks constructed by common components such as linear, polynomial, tangent hyperbolic, softplus, and sigmoid activation functions; squared, logistic, Huber, cross-entropy, and exponential loss functions satisfy the Łojasiewicz property [70, 76].

Next, we present the assumptions regarding the objective and constraint functions.

**Assumption 2.1** (Objective function). *(i) Function $\phi(\cdot, \cdot)$ is continuously differentiable and there exist constants $L_{xx}^{\phi}, L_{yy}^{\phi} \geq 0$ and $L_{xy}^{\phi} > 0$ such that for any $x, \bar{x} \in \mathbb{R}^n$ and $y, \bar{y} \in Y$,*

$$\|\nabla_x \phi(x, y) - \nabla_x \phi(\bar{x}, \bar{y})\| \leq L_{xx}^{\phi}\|x - \bar{x}\| + L_{xy}^{\phi}\|y - \bar{y}\|,$$
$$\|\nabla_y \phi(x, y) - \nabla_y \phi(\bar{x}, \bar{y})\| \leq L_{xy}^{\phi}\|x - \bar{x}\| + L_{yy}^{\phi}\|y - \bar{y}\|.$$

*(ii) $\nabla_x \phi$ is bounded, i.e., there exists $C_\phi > 0$ such that $\|\nabla_x \phi(x, y)\| \leq C_\phi$ for all $x \in \mathbb{R}^n$ and $y \in Y$. (iii) For any fixed $x \in \mathbb{R}^n$, $\phi(x, \cdot)$ is either $\eta_\phi$-strongly concave function or $c_\phi$-PL with $Y = \mathbb{R}^m$.*

**Assumption 2.2** (Constraint function). *(i) Function $\psi(\cdot, \cdot)$ is continuously differentiable and there exist constants $L_{xx}^{\psi}, L_{ww}^{\psi} \geq 0$ and $L_{xw}^{\psi} > 0$ such that for any $x, \bar{x} \in \mathbb{R}^m$ and $w, \bar{w} \in W$,*

$$\|\nabla_x \psi(x, w) - \nabla_x \psi(\bar{x}, \bar{w})\| \leq L_{xx}^{\psi}\|x - \bar{x}\| + L_{xw}^{\psi}\|w - \bar{w}\|,$$
$$\|\nabla_w \psi(x, w) - \nabla_w \psi(\bar{x}, \bar{w})\| \leq L_{xw}^{\psi}\|x - \bar{x}\| + L_{ww}^{\psi}\|w - \bar{w}\|.$$

*(ii) $\nabla_x \psi$ is bounded, i.e., there exists a constant $C_\psi > 0$ such that $\|\nabla_x \psi(x, w)\| \leq C_\psi$ for all $x \in \mathbb{R}^n$ and $w \in W$. (iii) For any fixed $x \in \mathbb{R}^n$, $\psi(x, \cdot)$ is either $\eta_\psi$-strongly concave function on a closed convex set $W \subseteq \mathbb{R}^\ell$ or $c_\psi$-PL over the entire space (i.e., $W = \mathbb{R}^\ell$).*

## 2.1 Regularity Assumption and Connection to Existing Literature

Finding a global/local solution of Nonlinear Programming (NLP) in (2) with nonconvex constraints is generally intractable. As such, most existing methods aim for finding a stationary solution known as the Krush-Kuhn-Tucker (KKT) point, i.e., finding $x \in X$ such that

$$0 \in \nabla f(x) + \mathbf{J}g(x)^\top \lambda + \mathcal{N}_X(x), \quad g(x) \leq \mathbf{0}, \quad \lambda_i g_i(x) = 0, \quad \forall i \tag{4}$$

for some $\lambda \in \mathbb{R}_+^m$. Even finding such a stationary solution is a daunting task, as one of the primary challenges lies in identifying a feasible solution when the constraint is nonconvex. In this setting, researchers have explored different assumptions and problem structures to ensure convergence of optimization algorithms to a feasible stationary point. For instance, assuming that the algorithm can start from a (nearly) feasible solution, several studies (e.g., [49, 40, 66]) have established convergence guarantees for obtaining an approximate KKT point. Another widely adopted assumption in the literature is a *regularity condition*, which posits the existence of a constant $\mu > 0$ such that $\|[g(x)]_+\| \leq \frac{\mu}{2}\text{dist}(\mathbf{J}g(x)^\top [g(x)]_+, -\mathcal{N}_X(x))$. In the special case where $X = \mathbb{R}^n$, this simplifies to

$$\|[g(x)]_+\| \leq \frac{\mu}{2}\|\mathbf{J}g(x)^\top [g(x)]_+\|. \tag{5}$$

Denoting the squared infeasibility residual function by $p(x) \triangleq \|[g(x)]_+\|^2$, this condition can be written equivalently as $p(x) \leq \mu^2\|\nabla p(x)\|^2$. Assuming that a feasible solution exists, we have $\min_x p(x) = 0$, making it clear that the above regularity condition is equivalent to $p(\cdot)$ satisfying the PL condition, that is, the Łojasiewicz inequality with exponent $\theta = 1/2$.

In this work, we consider a more general regularity condition based on the Łojasiewicz inequality (3) with an exponent $\theta \in (0, 1)$ for the residual function $[\psi(\cdot, w)]_+^2$ for any index parameter $w \in W$. Under this condition, we establish a uniform convergence guarantee for the proposed algorithm, which depends explicitly on the parameter $\theta$. We now formally introduce our regularity assumption.

**Assumption 2.3.** *Consider the constraint function $\psi : \mathbb{R}^n \times \mathbb{R}^\ell \to \mathbb{R}$ in problem (1). We assume that $[\psi(\cdot, w)]_+^2$ satisfies Łojasiewicz inequality for any given $w \in W$, i.e., there exist $\mu > 0$ and $\theta \in (0, 1)$ such that for any fixed $w \in W$, the following holds*

$$[\psi(x, w)]_+^{2\theta} \leq \mu\|\nabla_x \psi(x, w)[\psi(x, w)]_+\|, \quad \forall x \in \mathbb{R}^n. \tag{6}$$

*Remark* 2.1. We would like to highlight that this assumption generalizes existing regularity conditions in the nonconvex constrained optimization literature for any parameter $\theta \in (0,1)$. Notably, when the index set $W$ is a singleton and $\theta = 1/2$, this condition reduces to the classical regularity condition (5) for a single constraint. Furthermore, Assumption 2.3 is satisfied in a variety of practical applications, including the DRO-MTL described in Section 1.2. In particular, using results from the calculus of real analytical and semialgebraic functions [63, 35, 9], one can verify that Assumption 2.3 holds for $\psi(x,w) = \sum_j w_j \ell(x; \xi_j)$, where $\ell(\cdot; \xi)$ is a smooth loss function, such as those used in DNNs with smooth activation functions. However, Assumption 2.3 may not hold for neural networks with nonsmooth components such as ReLU, and one needs to use the smoothed variant, e.g., softplus, to ensure it holds.

## 3 Proposed Approach

In this paper, we introduce a novel dynamic barrier method tailored for min-max problems with semi-infinite, nonconvex constraints. Our proposed iterative scheme generates approximated gradient-based directions to update the triplet of variables $(x, y, w)$. Specifically, to minimize the objective function $\phi(x,y)$ with respect to $x$, we aim to follow a direction close to $\nabla_x \phi(x_k, y_k)$. Moreover, to encourage feasibility, this direction should either improve $\psi(x,w)$ in $x$ or avoid moving far away from the feasible region, i.e., $\max_{w \in W} \psi(x_{k+1}, w) \leq \max_{w \in W} \psi(x_k, w) + \varepsilon$ for a small and controllable error $\varepsilon > 0$. Indeed, we show that given a "good" estimate $w_k \approx w^*(x_k) \in \operatorname{argmax}_{w \in W} \psi(x_k, w)$, we can satisfy this condition by imposing an affine constraint $\nabla_x \psi(x_k, w_k)^\top d_k + \alpha_k \rho(x_k, w_k) \leq 0$, where $\rho(\cdot, \cdot)$ is an *inexact min-max dynamic barrier function*. Intuitively, function $\rho(\cdot, \cdot)$ encourages the search direction $d_k$ to align with $-\nabla_x \psi(x_k, w_k)$. Accordingly, we define $\rho(x_k, w_k) := \|\nabla_x \psi(x_k, w_k)\|$. The direction $d_k$ is then obtained by solving the following quadratic program (QP):

$$d_k = \operatorname{argmin}_d \|d + \nabla_x \phi(x_k, y_k)\|^2 \tag{7}$$
$$\text{s.t.} \qquad \nabla_x \psi(x_k, w_k)^\top d + \alpha_k \rho(x_k, w_k) \leq 0.$$

This QP admits a closed-form solution and can be computed efficiently at each iteration. More specifically, $d_k = -\nabla_x \phi(x_k, y_k) - \lambda_k \nabla_x \psi(x_k, w_k)$ where $\lambda_k$ is the dual multiplier corresponding to the constraint in the above QP updated as follows:

$$\lambda_k = \frac{1}{\|\nabla_x \psi(x_k, w_k)\|^2} [-\nabla_x \psi(x_k, w_k)^\top \nabla_x \phi(x_k, y_k) + \alpha_k \rho(x_k, w_k)]_+. \tag{8}$$

The main issue with the update of dual multiplier $\lambda_k$ is that its value goes to infinity as $\|\nabla_x \psi(x_k, w_k)\|$ vanishes. To resolve this issue, our idea is to introduce an indicator function $\zeta(x, w) \triangleq [\psi(x, w)]_+ \|\nabla_x \psi(x, w)\|$. Note that $\zeta(x, w^*(x)) = 0$ indicates that the point $x$ is feasible or a critical point of the constraint function. However, under Assumption 2.3 and using the definition of $\zeta(x, w) = \|\nabla_x \psi(x, w)[\psi(x, w)]_+\|$, we can conclude that $\zeta(x, w^*) = 0$ implies that $x$ is feasible. In this case, due to feasibility, we only wish to reduce the objective function and move along the direction $d_k = -\nabla_x \phi(x_k, y_k)$, hence, we would like to enforce $\lambda_k = 0$. However, computing the exact value of $\zeta(x_k, w^*(x_k))$ may not be possible. To resolve this issue, we use an estimated value $\zeta(x_k, w_k)$ as a measure of criticality and feasibility of the constraint which indicates whether $\lambda_k$ is updated based on (8) or set to zero. In other words, when $\zeta(x_k, w_k) = 0$ the constraints are treated as inactive. The maximization variables $y$ and $w$ are subsequently updated using $N_k$ and $M_k$ steps of (accelerated) gradient ascent, respectively. A full description of the algorithmic steps is presented in Algorithm 1.

*Remark* 3.1. We would like to point out that the proposed approach is related to the dynamic barrier gradient method introduced in [25]. While there are some conceptual similarities, we emphasize that the two methods differ significantly in both algorithmic design and convergence analysis. In particular, apart from addressing inexactness and incorporating a maximization component, two key distinctions of our approach are: (1) Introduction of the indicator function $\zeta(\cdot, \cdot)$, which serves as a metric to regulate the behavior of the dual multiplier $\lambda_k$. This leads to a modified update rule that enables convergence to a KKT point without requiring the boundedness of the dual iterates, unlike the assumption in [25]. (2) Our choice of barrier function differs: specifically, $\rho(x, w) = \|\nabla_x \psi(x, w)\|$ corresponds to $\tau = 1$ in their framework. However, this parameter choice does not yield a convergence rate guarantee in their method – see Proposition 3.7 in [25].

**Algorithm 1** Inexact Dynamic Barrier Primal-Dual (iDB-PD) Method for Semi-Infinite Min-Max
___
1: **Input:** $x_0 \in \mathbb{R}^n$, $\gamma_k \in (0,1)$, $\alpha_k > 0$, $N_k, M_k \geq 1$
2: **for** $k = 0, \ldots, T-1$ **do**
3:     $\lambda_k \leftarrow \begin{cases} \frac{1}{\|\nabla_x \psi(x_k, w_k)\|^2}[-\nabla_x \psi(x_k, w_k)^\top \nabla_x \phi(x_k, y_k) + \alpha_k \rho(x_k, w_k)]_+ & \text{if } \zeta(x_k, w_k) > 0 \\ 0 & \text{o.w.} \end{cases}$
4:     $d_k \leftarrow -\nabla_x \phi(x_k, y_k) - \lambda_k \nabla_x \psi(x_k, w_k)$
5:     $x_{k+1} \leftarrow x_k + \gamma_k d_k$
6:     $y_{k+1} \approx \operatorname{argmax}_{y \in Y} \phi(x_{k+1}, y)$    using $N_k$ steps of (accelerated) gradient ascent
7:     $w_{k+1} \approx \operatorname{argmax}_{w \in W} \psi(x_{k+1}, w)$    using $M_k$ steps of (accelerated) gradient ascent
8: **end for**
___

## 4   Convergence Analysis

In this section, we analyze the convergence of the iDB-PD algorithm. Indeed, problem (1) can be viewed as an implicit nonconvex constrained optimization problem $\min f(x)$ s.t. $g(x) \leq 0$, where $f(x) \triangleq \max_{y \in Y} \phi(x, y)$ and $g(x) \triangleq \max_{w \in W} \psi(x, w)$. Based on Assumption 2.1-(iii) and 2.2-(iii), we can show that $f$ and $g$ are continuously differentiable functions (see Section A.1 for proof); hence, we can establish the gap function based on the KKT solution of the implicit problem. Specifically, our goal is to find an $\epsilon$-KKT solution, i.e., find $\bar{x} \in \mathbb{R}^n$ and $\lambda \geq 0$ such that

$$\|\nabla f(\bar{x}) + \lambda \nabla g(\bar{x})\| \leq \epsilon, \quad [g(\bar{x})]_+ \leq \epsilon, \quad |\lambda g(\bar{x})| \leq \epsilon.$$

Our first step of analysis is to provide a descent-type inequality for the objective and constraint functions. This requires first analyzing the modified dual multiplier $\lambda_k$. The reason why we call it a modified dual multiplier is that $\lambda_k$-update is modified based on the value of the indicator function $\zeta(x_k, w_k)$. In effect, this modification can be translated into how we construct the QP subproblem. In particular, when $\zeta(x_k, w_k) > 0$, $d_k$ is updated based on the QP in (7), otherwise $d_k$ is updated based on the unconstrained variant of (7) , i.e., $d_k = \operatorname{argmin}_d \|d + \nabla_x \phi(x_k, y_k)\|^2$. Nevertheless, we can upper bound $\|d_k\|$ and $\lambda_k$. In particular, using Cauchy-Schwartz and triangle inequalities, one can verify that $\lambda_k \|\nabla_x \psi(x_k, w_k)\| \leq \|\nabla_x \phi(x_k, y_k)\| + \alpha_k$ for any $k \geq 0$. This relation, along with Lipschitz continuity of $\nabla f$ allows us to prove the following result regarding the objective function,

$$f(x_{k+1}) \leq f(x_k) + \gamma_k \alpha_k (C_\phi + \alpha_k) + \frac{L^\phi_{xy}}{2}\|y_k - y^*(x_k)\|^2 + \left(\frac{\gamma_k^2(L_f + L^\phi_{xy})}{2} - \gamma_k\right)\|d_k\|^2, \quad (9)$$

where $y^*(x) = \mathcal{P}_{Y^*(x)}(y_k)$ and $Y^*(x) = \operatorname{argmax}_{y \in Y} \phi(x, y)$.

To obtain a descent-type inequality for the constraint, we define the infeasibility residual function $p(x) \triangleq [g(x)]_+^2$. Note that this function is continuously differentiable whose gradient, i.e., $\nabla p(x) = 2\nabla g(x)[g(x)]_+$, is locally Lipschitz continuous with constant $L_p(x) \triangleq 2C_\psi^2 + L_g^2 + p(x)$ – see Lemma A.4 for details and proof. Although $d_k$ is not directly a feasible solution of the QP subproblem in (7) (it is only feasible if $\zeta(x_k, w_k) > 0$), we can show the following important inequality for $k \geq 0$,

$$[\psi(x_k, w_k)]_+ \nabla_x \psi(x_k, w_k)^\top d_k \leq -\alpha_k [\psi(x_k, w_k)]_+ \rho(x_k, w_k) = -\alpha_k \zeta(x_k, w_k).$$

Combining these results and conducting some extra analysis, we can show the following result regarding the infeasibility residual function:

$$
\begin{aligned}
p(x_{k+1}) \leq \; & p(x_k) - 2\gamma_k \alpha_k \zeta(x_k, w_k) + 2\gamma_k C_\psi |g(x_k) - \psi(x_k, w_k)| \|d_k\| \\
& + L^\psi_{xw} p(x_k)\|w_k - w^*(x_k)\|^2 + \frac{\gamma_k^2(L_p(x_k) + 2L^\psi_{xw})}{2}\|d_k\|^2,
\end{aligned} \quad (10)
$$

where $w^*(x) = \mathcal{P}_{W^*(x)}(w_k)$ and $W^*(x) = \operatorname{argmax}_{w \in W} \psi(x, w)$. The detailed statements and proof of the results in (9) and (10) are presented in Section A.3.

Although these results provide some bound on the progress of the next iterate with respect to objective and constraint functions, (10) may not immediately lead to a convergence rate result due to the dependencies of non-negative terms on the right-hand side to $p(x_k)$–especially notice the effect of

$p(x_k)$ in $L_p(x_k)$. To address this, we show that by carefully selecting the algorithm's parameters the sequence $\{\gamma_k\|d_k\|^2\}_{k\geq0}$ is summable and $\{p(x_k)\}_{k\geq0}$ is a bounded sequence – see Lemma A.7. As a side result, we can conclude that there exists a constant $L_p$ that can upper bound the local Lipschitz constant $L_p(x)$ uniformly along the sequence $\{x_k\}_{k\geq0}$ generated by Algorithm 1. Through this critical result, we can show the following theorem, establishing convergence bounds on $\|d_k\|^2$ and $[g(x_k)]_+$.

**Theorem 4.1.** *Suppose Assumptions 2.1, 2.2, and 2.3 hold. Let $\{x_k, \lambda_k\}_{k\geq0}$ be the sequence generated by Algorithm 1 such that $\{\alpha_k\}_k$ is a non-increasing sequence and $\gamma_k \leq (L_f + L_{xy}^\phi)^{-1}$. Then, for any $T \geq 1$,*

$$
\textbf{(I)} \quad \frac{1}{\Gamma_T}\sum_{k=0}^{T-1}\gamma_k\|d_k\|^2 \leq \frac{2(f(x_0)-f(x_T))}{\Gamma_T} + \frac{1}{\Gamma_T}\sum_{k=0}^{T-1}\gamma_k\alpha_k(C_\phi+\alpha_k) + \frac{1}{\Gamma_T}\sum_{k=0}^{T-1}\mathcal{E}_k^y, \quad (11)
$$

$$
\textbf{(II)} \quad \frac{1}{A_T}\sum_{k=0}^{T-1}\alpha_k[g(x_k)]_+^{2\theta} \leq \frac{\mu}{A_T}\sum_{k=0}^{T-1}\left(\frac{p(x_k)}{\gamma_k} - \frac{p(x_{k+1})}{\gamma_k}\right) + \frac{1}{A_T}\sum_{k=0}^{T-1}\mathcal{E}_k^w
$$

$$
+ \frac{\mu(L_p+2L_{xw}^\psi)}{2A_T}\sum_{k=0}^{T-1}\gamma_k\|d_k\|^2, \quad (12)
$$

*for some summable sequences $\{\mathcal{E}_k^y, \mathcal{E}_k^w\}_{k\geq0} \subset \mathbb{R}_+$, where $\Gamma_T \triangleq \sum_{k=0}^{T-1}\gamma_k$ and $A_T \triangleq \sum_{k=0}^{T-1}\alpha_k$.*

*Proof.* See Section A.4 in the Appendix. $\square$

Theorem 4.1 provides an upper bound on the accumulation of sequences of direction $d_k$ and infeasibility measure $[g(x_k)]_+$. While these quantities are not initially expressed in terms of KKT residuals, the following theorem establishes their connection to $\epsilon$-KKT conditions and derives the resulting complexity guarantees for the proposed algorithm.

**Theorem 4.2.** *Suppose Assumptions 2.1, 2.2, and 2.3 hold. Let $\{x_k, \lambda_k\}_{k=0}^{T-1}$ be the sequence generated by Algorithm 1 such that for any $k \geq 0$, $\alpha_k = \frac{T^{1/3}}{(k+2)^{1+\omega}}$, for some small $\omega > 0$, $\gamma_k = \gamma = \mathcal{O}(\min\{\frac{1}{T^{1/3}}, (L_f + L_{xy}^\phi)^{-1}\})$, $N_k = \mathcal{O}(\log(k+1))$, and $M_k = \mathcal{O}(\max\{\max\{1, \frac{1}{2\theta}\}\log(T), \log(T[\psi(x_k,w_k)]_+^{4\theta-2})\})$ if $\zeta(x_k, w_k) > 0$, otherwise, $M_k = \mathcal{O}(\max\{1, \frac{1}{2\theta}\}\log(T))$. Then, for any $\epsilon > 0$, there exists $t \in \{0, \dots, T-1\}$ such that*

1. *(Stationarity) $\|\nabla f(x_t) + \lambda_t \nabla g(x_t)\| \leq \epsilon$ within $T = \mathcal{O}(\frac{1}{\epsilon^3})$ iterations;*

2. *(Feasibility) $[g(x_t)]_+ \leq \epsilon$ within $T = \mathcal{O}(\frac{1}{\epsilon^{6\theta}})$ iterations;*

3. *(Slackness) $|\lambda_t g(x_t)| \leq \epsilon$ within $T = \mathcal{O}(\frac{1}{\epsilon^{3\theta/(1-\theta)}})$ iterations.*

*Proof.* See Section A.4 in the Appendix. $\square$

*Remark* 4.1. We would like to state some important remarks regarding the complexity result in the above theorem.

**(i) Convergence result:** We highlight that the obtained complexity results are, to the best of our knowledge, the first established non-asymptotic complexity bounds for nonconvex semi-infinite min-max problems. Owing to modest assumptions and a simple algorithmic structure, the proposed method is broadly applicable to problems, including those involving instances of DNNs.

**(ii) Special case of $\theta = \frac{1}{2}$:** The results of Theorem 4.2 simplify significantly when $\theta = \frac{1}{2}$, which corresponds to the PL condition on the squared inexact infeasibility residual $[\psi(x,w)]_+^2$ for any $w \in W$–see Assumption 2.3. In this case, we have $M_k = \mathcal{O}(\log(T))$, and the complexity for all three metrics improves to $\mathcal{O}(1/\epsilon^3)$, matching the best known results in [39, 47] for nonconvex optimization problems with finitely many constraints.

**(iii) Selection of $M_k$:** The choice of $M_k$ depends on the inexact infeasibility residual function when $\theta \in (0, 1)$, arising from the infinite cardinality of constraints. Here, $M_k$ controls the accuracy of estimating the iterate $w_k$ from this collection. As the iterates approach the approximate feasible region, increasingly accurate estimates are needed to ensure convergence to the true feasible region.

# 5 Numerical Experiments

In this section, we evaluate the performance of the proposed iDB-PD algorithm on the Robust Multi-Task 2 problem with a task priority, as introduced in Section 1.2. All experiments were implemented in PyTorch and executed on Google Colab, using a virtual machine equipped with an NVIDIA A100-SXM4 GPU (40 GB), an Intel® Xeon® CPU @ 2.20 GHz, 87 GB of RAM, and running Ubuntu 22.04.4 LTS with Python 3.12. We consider the following robust MTL formulation, where the goal is to learn two related tasks by optimizing a shared parameter vector. Specifically, the objective is to minimize the worst-case loss of a prioritized task while ensuring that the loss of the remaining task remains below a specified threshold denoted by $r > 0$, formulated as follows:

$$\min_{x \in \mathbb{R}^d} \max_{y \in \Delta_n} \quad \sum_{i=1}^{n} y_i \ell_1(x, \xi_i^{(1)}) \; - \; g_n(y) \tag{13a}$$

$$\text{s.t.} \quad \sum_{j=1}^{m} w_j \ell_2(x, \xi_j^{(2)}) \; - \; g_m(w) \leq r, \quad \forall w \in \Delta_m, \tag{13b}$$

where $\Delta_n$ and $\Delta_m$ are simplex sets. Note that, $g_n(y) = \frac{\lambda n}{2}\|y - \frac{1}{n}\mathbf{1}_n\|_2^2$ and $g_m(w) = \frac{\lambda m}{2}\|w - \frac{1}{m}\mathbf{1}_m\|_2^2$ are regularization terms that restrict the worst-case distributions from deviating significantly from the uniform distribution. Here, $\mathbf{1}$ denotes the all-ones vector. We consider five different datasets and, for each, evenly partition the labels into two disjoint subsets, and the goal of each task is to learn the corresponding labels. We consider a fully connected neural network with one hidden layer and tanh activation functions. The output layer is a softmax with a cross-entropy loss function $\ell_i(\cdot)$.

Our experiment includes five datasets as summarized in Table 1. We used Multi-MNIST and Multi-Fashion-MNIST from Lin et al. (2019) [44][1], which were constructed from the original MNIST dataset. For these datasets, each data point is constructed by randomly sampling two different images from the original MNIST (Fashion-MNIST) dataset and combining them into a single image, placing the two digits (or

Table 1: Summary of datasets used in the experiments.

| Dataset | Instances | Features | Labels |
|---|---|---|---|
| Multi-MNIST | 20000 | 1296 | 10 |
| CHD49 | 555 | 49 | 6 |
| Multi-Fashion MNIST | 20000 | 1296 | 10 |
| Yeast | 2417 | 103 | 14 |
| 20NG | 19300 | 1006 | 20 |

articles of clothing) on the top left and bottom right corners, resulting in a $36 \times 36$ image. Yeast, Coronary Heart Disease, and 20NewsGroup datasets were accessed via the Multi-Label Classification Dataset Repository hosted by Universidad de Córdoba[2]. For each of these datasets, we evenly divide the label set in two, with one half learned in the objective and the other half learned in the constraint. Here, we report our results on Multi-MNIST and CHD49, while results on the remaining datasets (Multi-Fashion MNIST, Yeast, and 20NG) are included in Appendix A.5.

**Experiment 1.** In this experiment, we compare iDB-PD with an adaptive discretization method, which iteratively adds the most violated constraints to form a finite approximation of the semi-infinite min–max problem following Blankenship and Falk [8]. The resulting discretized problem is then solved by COOPER [24], a PyTorch library for constrained optimization that implements first-order, Lagrangian-based update schemes. From Figure 1 we observe that our proposed method (iDB-PD) consistently achieves convergence in infeasibility, stationarity, and slackness, whereas the adaptive discretization fails to reduce infeasibility and diverges in stationarity. This result underscores the advantage of iDB-PD over more classical discretization methods.

---

[1]Datasets by Lin et al. (2019): https://github.com/Xi-L/ParetoMTL/
[2]https://www.uco.es/kdis/mllresources/

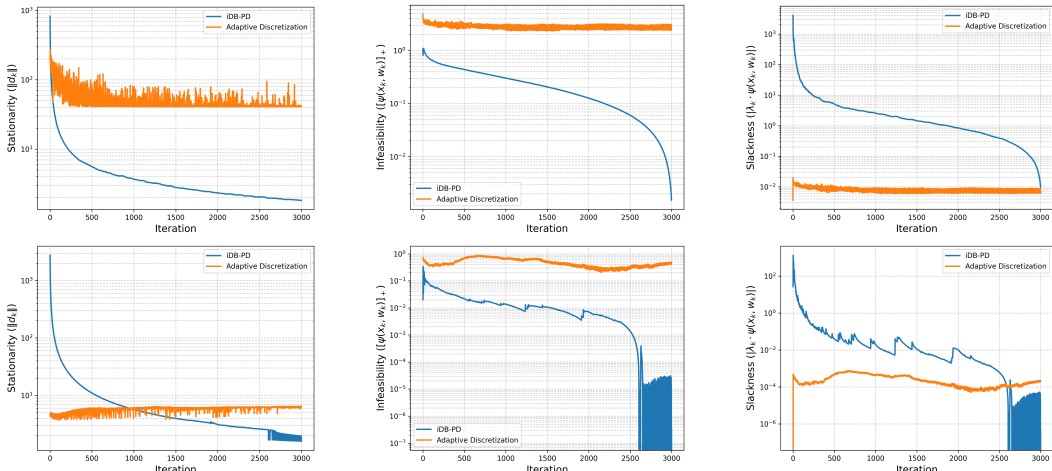

Figure 1: iDB-PD vs. Adaptive Discretization with COOPER on multi-MNIST (top row) and CHD49 (bottom row), evaluated in terms of stationarity, infeasibility, and slackness.

**Experiment 2.** In this experiment, we compare the performance of the model in (13) with an alternative model in which the objective function is a weighted summation of loss functions with a DRO formulation. This results in the following min-max problem formulation

$$
\min_{x \in \mathbb{R}^d} \max_{y \in \Delta_n, w \in \Delta_m} \sum_{i=1}^{n} y_i \ell_1(x, \xi_i^{(1)}) - g_n(y) + \rho \left( \sum_{j=1}^{m} w_j \ell_2(x, \xi_j^{(2)}) - g_m(w) \right).
$$

For a fair comparison, we applied the Gradient Descent Multi-Ascent (GDMA) method [31, 57] to solve this reformulated min-max problem, testing several values $\rho \in \{1, 2, 5, 10\}$. We evaluated both approaches in terms of the metrics corresponding to (13), i.e., the objective function value, infeasibility, and stationarity. Note that the first two metrics correspond to the training losses of task 1 and task 2, respectively.

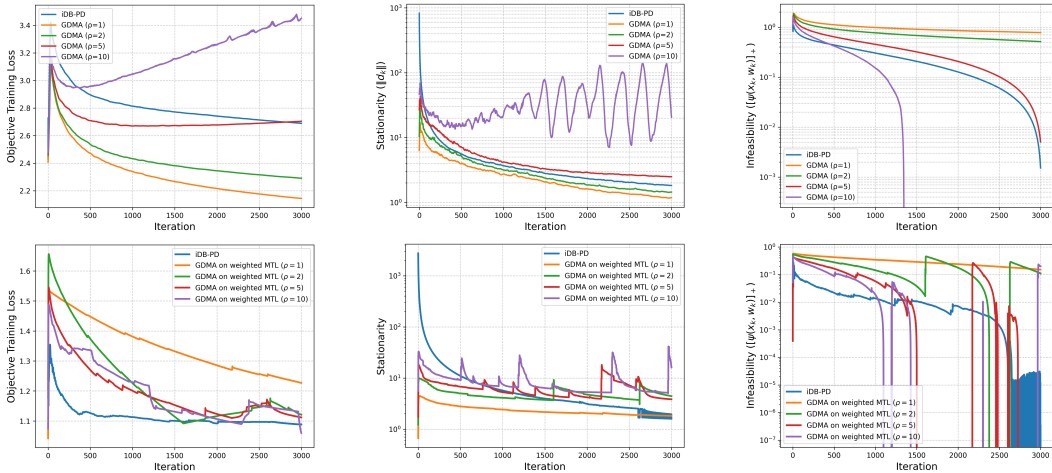

Figure 2: iDB–PD vs GDMA on multi-MNIST (top row) and CHD49 (bottom row), evaluated in terms of stationarity, infeasibility, and objective loss.

In Figure 2, we observe that iDB-PD consistently converges in both feasibility and stationarity, whereas GDMA with small $\rho$ achieves low stationarity, but fails to reduce infeasibility, and GDMA with large $\rho$ exhibits unstable behavior. These results highlight the difficulty of selecting appropriate loss weights ($\rho$) to balance the tasks and confirm the robustness of our semi-infinite constrained min-max formulation, where such a weight is dynamically adjusted by the algorithm.

## Acknowledgments and Disclosure of Funding

This work is supported in part by the National Science Foundation under Grants ECCS-2515979 and 2231863.

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

# A   Technical Appendices and Supplementary Material

## A.1   Analysis of the maximization variables

Consider the following parametric maximization problem

$$\tilde{h}^*(x) \triangleq \max_{u \in U} h(x, u), \tag{14}$$

for a given $x \in \mathbb{R}^n$ where $h(\cdot, \cdot) : \mathbb{R}^n \times \mathbb{R}^m \to \mathbb{R}$ is a continuously differentiable function, and $U \subseteq \mathbb{R}^m$ is a closed, convex set. We are interested in the conditions under which the value function $\tilde{h}^*(x)$ is Lipschitz differentiable and an approximate solution of the above problem can be found using (accelerated) gradient ascent method. Indeed, using the classical result in the optimization literature [32, 56, 55, 57], it can be deduced that both these properties are satisfied when $h(x, \cdot)$ is strongly concave or satisfies PL inequality when $U = \mathbb{R}^m$. In the following, we will restate these results in a unified statement and later specify them for our proposed algorithm. In particular, we first state the linear convergence result under these conditions, and then state the differentiability of the value function.

**Proposition A.1** ([32, 56, 55]). *Consider problem* (14)*, and assume that $h(\cdot, \cdot)$ is a continuously differentiable function, such that for any fixed $x$, $h(x, \cdot)$ is either strongly concave (or satisfies PL inequality (see Def. 3 with $\theta = \frac{1}{2}$) with $U = \mathbb{R}^m$). Let $\{u_k\}_{k=0}^{T-1} \subset \mathbb{R}^m$ be a sequence generated by the (accelerated) gradient ascent method. Then, for any $x \in \mathbb{R}^n$, there exists $\delta \in (0, 1)$ and $\Delta_1, \Delta_2 > 0$ such that we have the following results for $T \geq 1$,*

$$\|u_T - u^*(x)\|^2 \leq \Delta_1 \delta^T,$$
$$h(x, u_T) \leq \tilde{h}^*(x) \leq h(x, u_T) + \Delta_2 \delta^T,$$

*where $u^*(x) \triangleq \mathcal{P}_{U^*(x)}(u_T)$, and $U^*(x) \triangleq \operatorname{argmax}_{u \in U} h(x, u)$.*

**Proposition A.2** ([57] Lemma A.5). *Consider problem* (14)*, and assume that $h(\cdot, \cdot)$ is a continuously differentiable function, such that for any fixed $x$, $h(x, \cdot)$ is $\eta_h$-strongly concave (or $c_h$-PL (see Def. 2.1 with $\theta = \frac{1}{2}$) with $U = \mathbb{R}^m$), it follows that*

$$\nabla \tilde{h}^*(x) = \nabla_x h(x, u^*) \quad \text{for any } u^* \in U^*(x).$$

*where $U^*(x) \triangleq \operatorname{argmax}_{u \in U} h(x, u)$. Moreover, $\tilde{h}^*(x)$ has $(L_{uu}^h + (L_{xu}^h)^2/\iota)$-Lipschitz gradient where $\iota = \eta_h$ (or $\iota = c_h^2$).*

Now, we apply the above propositions to the objective and constraint functions in problem (1) to derive the error of estimating the maximization components according to the updates of Algorithm 1, which will be used in the analysis. Recall that $f(x) = \max_{y \in Y} \phi(x, y)$ and $g(x) = \max_{w \in W} \psi(x, w)$. Based on Proposition A.2 and Assumptions 2.1 and 2.2, we have the following properties:

1. $f$ is continuously differentiable and has a Lipschitz gradient with constant $L_f \triangleq L_{yy}^\phi + (L_{xy}^\phi)^2/\iota_f$,

2. $g$ is continuously differentiable and has a Lipschitz gradient with constant $L_g \triangleq L_{ww}^\psi + (L_{xw}^\psi)^2/\iota_g$,

where $\iota_f = \eta_\psi$ when $\psi(x, \cdot)$ is $\eta_\psi$-strongly concave or $\iota_f = c_\psi^2$ when $\psi(x, \cdot)$ is $c_\psi$-PL ($\iota_g$ is defined similarly).

Moreover, based on Proposition A.1, there exist uniform constants $\Delta_1^y, \Delta_1^w, \Delta_2^w \in (0, +\infty)$ and $\delta_y, \delta_w \in (0, 1)$, such that for any $k \geq 0$,

$$\|y_k - y^*(x_k)\|^2 \leq \Delta_1^y \delta_y^{N_k}, \tag{15}$$
$$\|w_k - w^*(x_k)\|^2 \leq \Delta_1^w \delta_w^{M_k}, \tag{16}$$
$$\psi(x_k, w_k) \leq g(x_k) \leq \psi(x_k, w_k) + \Delta_2^w \delta_w^{M_k}, \tag{17}$$

where $N_k, M_k$ denote the number of (accelerated) gradient ascent steps to maximize the functions $\phi(x, \cdot)$ and $\psi(x, \cdot)$, respectively.

## A.2 Required Lemmas

This section states two important lemmas regarding the proposed method and the implicit constraint function. First, we show some bounds on the modified dual multiplier $\lambda_k$ corresponding to the subproblem (7) based on the update of Algorithm 1. Next, we establish local Lipschitz continuity of the infeasibility residual function $p(x) \triangleq [g(x)]_+^2$. Later, we show that by carefully selecting the stepsize, this local constant can be upper bounded by a global one.

**Lemma A.3.** *Suppose Assumptions 2.1 and 2.2 hold, and let $\{x_k, \lambda_k\}_{k \geq 0}$ be the sequence generated by Algorithm 1 such that $\{\alpha_k\}_{k \geq 0} \subset \mathbb{R}_+$ is non-increasing sequence. Then, for any $k \geq 0$ we have that $\lambda_k \rho(x_k, w_k) \leq \|\nabla_x \phi(x_k, y_k)\| + \alpha_k$. Furthermore, if Assumption 2.3 hold, then for any $k \geq 0$, $\lambda_k [g(x_k)]_+ \leq C[g(x_k)]_+^{2-2\theta} + C\Delta_2^w \delta_w^{M_k} [\psi(x_k, w_k)]_+^{1-2\theta}$ where $C \triangleq \mu(C_\phi + \alpha_0)$.*

*Proof.* Recall that $\rho(x_k, w_k) = \|\nabla_x \psi(x_k, w_k)\|$ and $\zeta(x_k, w_k) = [\psi(x_k, w_k)]_+ \|\nabla_x \psi(x_k, w_k)\|$. Note that if $\lambda_k = 0$, the bound holds trivially. Now suppose, $\zeta(x_k, w_k) > 0$, then using the update of $\lambda_k$ we have that

$$\lambda_k \rho(x_k, w_k) = \frac{1}{\|\nabla_x \psi(x_k, w_k)\|} \left[ -\nabla_x \psi(x_k, w_k)^\top \nabla_x \phi(x_k, y_k) + \alpha_k \rho(x_k, w_k) \right]_+.$$

Taking the absolute value from both sides, using the fact that $|\max\{a, b\}| \leq |a| + |b|$ for any $a, b \in \mathbb{R}$, followed by the triangle and Cauchy-Schwarz inequalities, we conclude that $\lambda_k \|\nabla_x \psi(x_k, y_k)\| \leq \|\nabla_x \phi(x_k, y_k)\| + \alpha_k$.

Similarly, from the definition of $\lambda_k$ and Assumption 2.3 we conclude that

$$
\begin{aligned}
\lambda_k [g(x_k)]_+ &\leq \frac{[g(x_k)]_+}{\|\nabla_x \psi(x_k, w_k)\|} (\|\nabla_x \phi(x_k, y_k)\| + \alpha_k) \\
&\leq \mu [g(x_k)]_+ [\psi(x_k, w_k)]_+^{1-2\theta} (\|\nabla_x \phi(x_k, y_k)\| + \alpha_k) \\
&\leq \mu \left( [\psi(x_k, w_k)]_+^{2-2\theta} + \Delta_2^w \delta_w^{M_k} [\psi(x_k, w_k)]_+^{1-2\theta} \right) (\|\nabla_x \phi(x_k, y_k)\| + \alpha_0) \\
&\leq \mu \left( [g(x_k)]_+^{2-2\theta} + \Delta_2^w \delta_w^{M_k} [\psi(x_k, w_k)]_+^{1-2\theta} \right) (\|\nabla_x \phi(x_k, y_k)\| + \alpha_0)
\end{aligned}
$$

where in the penultimate inequality we used the second inequality in (17) and that $\alpha_k$ is a non-increasing sequence. The last inequality above follows from the first inequality in (17). Finally, the result follows from the boundedness of $\nabla_x \phi(\cdot, \cdot)$ – see Assumption 2.1. $\square$

**Lemma A.4.** *Suppose Assumption 2.2 holds. Let $g(x) \triangleq \max_{w \in W} \psi(x, w)$ and the infeasibility residual by $p(x) = [g(x)]_+^2$. Then $p(\cdot)$ is a continuously differentiable function and $\nabla p(x)$ is locally Lipschitz continuous with constant $L_p(x) \triangleq 2C_\psi^2 + L_g^2 + [g(x)]_+^2$.*

*Proof.* Differentiability of $p(\cdot)$ follows from differentiability of $g$ as established in Property 2 and its gradient can be calculated by the chain rule as $\nabla p(x) = 2\nabla g(x)[g(x)]_+$. Therefore, we have that

$$
\begin{aligned}
\|\nabla p(x) - \nabla p(y)\| &= \|2[g(x)]_+ \nabla g(x) - 2[g(y)]_+ \nabla g(y)\| \\
&= \|2[g(x)]_+ (\nabla g(x) - \nabla g(y)) + 2\nabla g(y)([g(x)]_+ - [g(y)]_+)\| \\
&\leq 2\|\nabla g(x) - \nabla g(y)\| \|[g(x)]_+\| + 2\|\nabla g(y)\| \|[g(x)]_+ - [g(y)]_+\|,
\end{aligned}
$$

where in the second equality we added and subtracted $2[g(x)]_+ \nabla g(y)$. Note that based on Assumption 2.2-(ii), we have that $\nabla g(x) = \nabla_x \psi(x, w^*(x))$ is bounded by $C_\psi$, hence, $g$ is $C_\psi$-Lipschitz continuous. Therefore,

$$\|\nabla p(x) - \nabla p(y)\| \leq \left( 2L_g [g(x)]_+ + 2C_\psi^2 \right) \|x - y\|.$$

From Young's inequality, we can bound $2L_g [g(x)]_+ \leq L_g^2 + [g(x)]_+^2$. Therefore, the following holds

$$\|\nabla p(x) - \nabla p(y)\| \leq (2C_\psi^2 + L_g^2 + [g(x)]_+^2)\|x - y\|.$$

$\square$

## A.3 Proof of one-step analysis

In this section, we prove the one-step analysis for the objective and constraints.

**Lemma A.5.** *Suppose Assumptions 2.1, 2.2, and 2.3 hold. Let $\{x_k, \lambda_k\}_{k\geq 0}$ be the sequence generated by Algorithm 1 such that $\{\alpha_k\}_k$ is a non-increasing sequence and $\gamma_k \leq (L_f + L_{xy}^\phi)^{-1}$. Then, for any $k \geq 0$*

$$\textbf{(I)} \quad \frac{\gamma_k}{2}\|d_k\|^2 \leq f(x_k) - f(x_{k+1}) + \gamma_k \alpha_k (C_\phi + \alpha_k) + \frac{L_{xy}^\phi}{2}\Delta_1^y \delta_y^{N_k}, \tag{18}$$

$$\textbf{(II)} \quad \gamma_k \alpha_k \zeta(x_k, w_k) \leq \frac{p(x_k)}{2} - \frac{p(x_{k+1})}{2} + \gamma_k \Delta_1^w \delta_w^{M_k} C_\psi (2C_\phi + \alpha_0)$$

$$+ \frac{L_{xw}^\psi}{2}p(x_k)\Delta_1^w \delta_w^{M_k} + \frac{\gamma_k^2(L_p(x_k) + 2L_{xw}^\psi)}{4}\|d_k\|^2. \tag{19}$$

*Proof.* **Part (I):** Using Lipschitz continuity of gradient of $f$ as established in Property 1 and update of $x_{k+1}$, we have that

$$f(x_{k+1}) = f(x_k) + \nabla f(x_k)^\top (x_{k+1} - x_k) + \frac{L_f}{2}\|x_{k+1} - x_k\|^2$$

$$= f(x_k) + \gamma_k \nabla f(x_k)^\top d_k + \frac{\gamma_k^2 L_f}{2}\|d_k\|^2$$

$$= f(x_k) + \gamma_k \left(\nabla_x \phi(x_k, y_k) + d_k\right)^\top d_k + \left(\frac{\gamma_k^2 L_f}{2} - \gamma_k\right)\|d_k\|^2$$

$$+ \gamma_k \|\nabla f(x_k) - \nabla_x \phi(x_k, y_k)\|\|d_k\|$$

$$\leq f(x_k) - \gamma_k \lambda_k \nabla_x \psi(x_k, w_k)^\top d_k + \left(\frac{\gamma_k^2 L_f}{2} - \gamma_k\right)\|d_k\|^2 + \gamma_k L_{xy}^\phi \|y_k - y^*(x_k)\|\|d_k\|,$$

where in the last inequality we used $d_k = -\nabla_x \phi(x_k, y_k) - \lambda_k \nabla_x \psi(x_k, w_k)$, $\nabla f(x_k) = \nabla_x \phi(x_k, y^*(x_k))$, and Lipschitz continuity of the gradient of function $\phi$. Moreover, from complementarity slackness condition we know that $\lambda_k \left(\nabla_x \psi(x_k, w_k)^\top d_k + \alpha_k \rho(x_k, w_k)\right) = 0$, hence we obtain

$$f(x_{k+1}) - f(x_k)$$

$$\leq \left(\frac{\gamma_k^2 L_f}{2} - \gamma_k\right)\|d_k\|^2 + \gamma_k \alpha_k \lambda_k \rho(x_k, w_k) + \gamma_k L_{xy}^\phi \|y_k - y^*(x_k)\|\|d_k\|$$

$$\leq \left(\frac{\gamma_k^2 L_f}{2} - \gamma_k\right)\|d_k\|^2 + \gamma_k \alpha_k (C_\phi + \alpha_k) + \gamma_k L_{xy}^\phi \|y_k - y^*(x_k)\|\|d_k\|$$

$$\leq \left(\frac{\gamma_k^2 L_f}{2} - \gamma_k\right)\|d_k\|^2 + \gamma_k \alpha_k (C_\phi + \alpha_k) + \frac{L_{xy}^\phi}{2}\|y_k - y^*(x_k)\|^2 + \frac{\gamma_k^2 L_{xy}^\phi}{2}\|d_k\|^2. \tag{20}$$

where the penultimate inequality follows from the application of Lemma A.3 and $\|\nabla_x \phi(x, y)\| \leq C_\phi$, moreover, the last inequality is due to Young's inequality (where $p = q = 2$). Now, rearranging the terms and selecting $\gamma_k \leq (L_f + L_{xy}^\phi)^{-1}$ lead to the result of part (I).

**Part (II):** Recall that $\zeta(x_k, w_k) = [\psi(x_k, w_k)]_+\|\nabla_x \psi(x_k, w_k)\|$ and $\rho(x_k, w_k) = \|\nabla_x \psi(x_k, w_k)\|$. Based on Lemma A.4 and the update rule of $x_{k+1} = x_k + \gamma_k d_k$, we have that

$$p(x_{k+1}) - p(x_k)$$

$$\leq \langle \nabla p(x_k), x_{k+1} - x_k \rangle + \frac{L_p(x_k)}{2}\|x_{k+1} - x_k\|^2$$

$$= 2\gamma_k[g(x_k)]_+ \nabla g(x_k)^\top d_k + \frac{\gamma_k^2 L_p(x_k)}{2}\|d_k\|^2$$

$$= 2\gamma_k[g(x_k)]_+ \nabla_x \psi(x_k, w_k)^\top d_k + 2\gamma_k[g(x_k)]_+(\nabla g(x_k) - \nabla_x \psi(x_k, w_k))^\top d_k + \frac{\gamma_k^2 L_p(x_k)}{2}\|d_k\|^2$$

$$\leq 2\gamma_k \underbrace{[g(x_k)]_+ \nabla_x \psi(x_k, w_k)^\top d_k}_{\text{term (a)}} + 2\gamma_k L_{xy}^\psi[g(x_k)]_+\|w_k - w^*(x_k)\|\|d_k\| + \frac{\gamma_k^2 L_p(x_k)}{2}\|d_k\|^2.$$

$$\tag{21}$$

Considering term (a), from (17) one can observe that

$$
\begin{aligned}
[g(x_k)]_+ \nabla_x \psi(x_k, w_k)^\top d_k &\leq [\psi(x_k, w_k)]_+ \nabla_x \psi(x_k, w_k)^\top d_k + \Delta_1^w \delta_w^{M_k} \|\nabla_x \psi(x_k, w_k)\| \|d_k\| \\
&\leq -\alpha_k [\psi(x_k, w_k)]_+ \rho(x_k, w_k) + \Delta_1^w \delta_w^{M_k} \|\nabla_x \psi(x_k, w_k)\| \|d_k\| \\
&\leq -\alpha_k \zeta(x_k, w_k) + \Delta_1^w \delta_w^{M_k} C_\psi (2C_\phi + \alpha_0), \quad (22)
\end{aligned}
$$

where in the second inequality we use the fact that $d_k$ is a feasible solution of the QP subproblem if $\zeta(x_k, w_k) > 0$, hence, $[\psi(x_k, w_k)]_+ \nabla_x \psi(x_k, w_k)^\top d_k \leq -\alpha_k [\psi(x_k, w_k)]_+ \rho(x_k, w_k) = -\alpha_k \zeta(x_k, w_k)$, otherwise the inequality holds trivially. Moreover, the last inequality follows from Assumption 2.1-(ii) and Lemma A.3 and one can easily verify that $\|d_k\| \leq 2C_\phi + \alpha_0$ and from Assumption 2.2 we have $\|\nabla_x \psi(x_k, w_k)\| \leq C_\psi$. Therefore, combining (22) with (21), we obtain

$$
\begin{aligned}
p(x_{k+1}) - p(x_k) &\leq -2\gamma_k \alpha_k \zeta(x_k, w_k) + 2\gamma_k \Delta_1^w \delta_w^{M_k} C_\psi (2C_\phi + \alpha_0) \\
&\quad + 2\gamma_k L_{xy}^\psi [g(x_k)]_+ \|w_k - w^*(x_k)\| \|d_k\| + \frac{\gamma_k^2 L_p(x_k)}{2} \|d_k\|^2 \\
&\leq -2\gamma_k \alpha_k \zeta(x_k, w_k) + 2\gamma_k \Delta_1^w \delta_w^{M_k} C_\psi (2C_\phi + \alpha_0) \\
&\quad + L_{xw}^\psi p(x_k) \|w_k - w^*(x_k)\|^2 + \frac{\gamma_k^2 (L_p(x_k) + 2L_{xw}^\psi)}{2} \|d_k\|^2.
\end{aligned}
$$

Next, rearranging the above inequality, dividing both sides by 2, lead to the desired result. $\qquad \square$

## A.4 Proof of Theorems 4.1 and 4.2

Before proving Theorems 4.1 and 4.2, we present a technical lemma on the recursive relation of a non-negative real-valued sequence that will be used in our convergence analysis.

**Lemma A.6** ([3] Lemma 5.31). *Let $\{v_k\}$, $\{u_k\}$, $\{\alpha_k\}$, $\{\beta_k\}$ be sequences of nonnegative reals with $\sum_{k=0}^\infty \alpha_k < \infty$ and $\sum_{k=0}^\infty \beta_k < \infty$ such that $v_{k+1} \leq (1 + \alpha_k)v_k - u_k + \beta_k$ for all $k$. Then, $\{v_k\}$ converges and $\sum_{k=0}^\infty u_k < \infty$.*

Using this result, we first show that the sequence $\{\gamma_k \|d_k\|^2\}_k$ is summable and $\{p(x_k)\}_k$ is a bounded sequence.

**Lemma A.7.** *Let $\{x_k\}_k$ be the sequence generated by Algorithm 1 such that $\sum_{k=0}^{+\infty} \gamma_k \alpha_k < +\infty$, $\sum_{k=0}^{+\infty} \delta_y^{N_k} < +\infty$, and $\sum_{k=0}^{+\infty} \delta_w^{M_k} < +\infty$. Under the premises of Lemma A.5, we have that (i) $\sum_{k=0}^{+\infty} \gamma_k \|d_k\|^2 < +\infty$; (ii) $\{p(x_k)\}_{k\geq 0}$ is a bounded sequence, i.e., there exists $C_g > 0$ such that $[g(x_k)]_+ \leq C_g$ for any $k \geq 0$.*

*Proof.* (i) Consider **Part (I)** of Lemma A.5 by rearranging terms one can obtain:

$$
f(x_{k+1}) \leq f(x_k) - \frac{\gamma_k}{2} \|d_k\|^2 + \gamma_k \alpha_k (C_\phi + \alpha_k) + \frac{L_{xy}^\phi}{2} \Delta_1^y \delta_y^{N_k}.
$$

Since $\alpha_k$ is a non-increasing sequence and it is assumed that $\sum_{k=0}^{+\infty} \gamma_k \alpha_k < +\infty$, one can verify that $\sum_{k=0}^{+\infty} \gamma_k \alpha_k^2 \leq \sum_{k=0}^{+\infty} \gamma_k \alpha_k < +\infty$. Moreover, since $\sum_{k=0}^{+\infty} \delta_y^{N_k} < +\infty$, we have $\sum_{k=0}^{+\infty} \left( \gamma_k \alpha_k (C_\phi + \alpha_k) + \frac{L_{xy}^\phi}{2} \Delta_1^y \delta_y^{N_k} \right) < +\infty$. Therefore, applying Lemma A.6, we conclude that $\sum_{k=0}^{+\infty} \gamma_k \|d_k\|^2 < +\infty$.

(ii) Similarly, from **Part (II)** of Lemma A.5, multiplying both sides by 2, using $L_p(x) = 2C_\psi^2 + L_g^2 + p(x)$ from Lemma A.4, and rearranging terms yields:

$$
\begin{aligned}
p(x_{k+1}) \leq \underbrace{(1 + L_{xw}^\psi \Delta_1^w \delta_w^{M_k} + \frac{\gamma_k^2}{2} \|d_k\|^2)}_{\mathbf{a}_k} p(x_k) - 2\gamma_k \alpha_k \zeta(x_k, w_k) \\
+ \underbrace{2\gamma_k \Delta_1^w \delta_w^{M_k} C_\psi (2C_\phi + \alpha_0) + \frac{\gamma_k^2 (2C_\psi^2 + L_g^2 + 2L_{xw}^\psi)}{2} \|d_k\|^2}_{\mathbf{b}_k}.
\end{aligned}
$$

From the assumptions in the statement of the lemma and the result of part (I) and that $\gamma_k \in (0,1)$, we have that $\sum_{k=0}^{+\infty} \mathbf{a}_k < +\infty$ and $\sum_{k=0}^{+\infty} \mathbf{b}_k < +\infty$. Hence, the conditions of Lemma A.6 are satisfied, and we conclude that the sequence $\{p(x_k)\}_{k\geq 0}$ converges. Therefore, $\{p(x_k)\}_{k\geq 0}$ is bounded, i.e., there exists $C_g > 0$ such that $[g(x_k)]_+ \leq C_g$ for all $k \geq 0$. $\qquad\square$

Now, we are ready to prove Theorem 4.1. First, we restate the statement with full details here.

**Theorem A.8** (Restatement of Theorem 4.1). *Suppose Assumptions 2.1, 2.2, and 2.3 hold. Let $\{x_k, \lambda_k\}_{k\geq 0}$ be the sequence generated by Algorithm 1 such that $\{\alpha_k\}_k$ is a non-increasing sequence and $\gamma_k \leq (L_f + L_{xy}^\phi)^{-1}$. Then, for any $T \geq 1$ and $k \geq 1$,*

*(I)* 
$$\frac{1}{\Gamma_T}\sum_{k=0}^{T-1}\gamma_k\|d_k\|^2 \leq \frac{2(f(x_0) - f(x_T))}{\Gamma_T} + \frac{1}{\Gamma_T}\sum_{k=0}^{T-1}\gamma_k\alpha_k(C_\phi + \alpha_k) + \frac{L_{xy}^\phi\Delta_1^y}{\Gamma_T}\sum_{k=0}^{T-1}\delta_y^{N_k},$$
(23)

*(II)*
$$\frac{1}{A_T}\sum_{k=0}^{T-1}\alpha_k[g(x_k)]_+^{2\theta} \leq \frac{\mu}{A_T}\sum_{k=0}^{T-1}\left(\frac{p(x_k)}{\gamma_k} - \frac{p(x_{k+1})}{\gamma_k}\right) + \frac{\mu}{A_T}\sum_{k=0}^{T-1}\left(\frac{\bar{\Delta}}{\gamma_k}\delta_w^{M_k} + \frac{2\alpha_k}{\mu}(\Delta_2^w)^{2\theta}\delta_w^{2\theta M_k}\right)$$
$$+ \frac{\mu(L_p + 2L_{xw}^\psi)}{2A_T}\sum_{k=0}^{T-1}\gamma_k\|d_k\|^2,$$
(24)

*for some $\bar{\Delta} > 0$, where $\Gamma_T \triangleq \sum_{k=0}^{T-1}\gamma_k$ and $A_T \triangleq \sum_{k=0}^{T-1}\alpha_k$.*

*Proof.* Part (I) follows immediately from Lemma A.5-Part (I) by summing over $k = 0$ to $T - 1$ and dividing both sides by $\Gamma_k = \sum_{k=0}^{T-1}\gamma_k$.

To prove Part (II), first note that from Lemma A.7 we have $[g(x_k)]_+ \leq C_g$ which from Lemma A.4 we conclude that there exists a constant $L_p \triangleq 2C_\psi^2 + L_g^2 + C_g^2$ that upper bounds the local Lipschitz constant $L_p(x)$ uniformly along the sequence $\{x_k\}_{k\geq 0}$. Therefore, we can simplify the bound in (19) as follows

$$\gamma_k\alpha_k\zeta(x_k, w_k) \leq \frac{p(x_k)}{2} - \frac{p(x_{k+1})}{2} + \gamma_k\Delta_1^w\delta_w^{M_k}C_\psi(2C_\phi + \alpha_0)$$
$$+ \frac{L_{xw}^\psi}{2}C_g^2\Delta_1^w\delta_w^{M_k} + \frac{\gamma_k^2(L_p + 2L_{xw}^\psi)}{4}\|d_k\|^2.$$

Using Assumption 2.3, we can lower bound the left-hand side of the above inequality by $\frac{\gamma_k\alpha_k}{\mu}[\psi(x_k, w_k)]_+^{2\theta}$. Moreover, from (17) and that $\theta \in (0,1)$ we have that $\frac{1}{2}[g(x_k)]_+^{2\theta} \leq [\psi(x_k, w_k)]_+^{2\theta} + (\Delta_2^w)^{2\theta}\delta_w^{2\theta M_k}$ which leads to

$$\frac{\gamma_k\alpha_k}{2\mu}[g(x_k)]_+^{2\theta} \leq \frac{p(x_k)}{2} - \frac{p(x_{k+1})}{2} + \gamma_k\Delta_1^w\delta_w^{M_k}C_\psi(2C_\phi + \alpha_0) + \frac{L_{xw}^\psi}{2}C_g^2\Delta_1^w\delta_w^{M_k}$$
$$+ \frac{\gamma_k^2(L_p + 2L_{xw}^\psi)}{4}\|d_k\|^2 + \frac{\gamma_k\alpha_k}{\mu}(\Delta_2^w)^{2\theta}\delta_w^{2\theta M_k}.$$

Finally, multiplying both sides by $2\mu/\gamma_k$, summing over $k = 0$ to $T - 1$, dividing by $A_T$, and defining $\bar{\Delta} \triangleq \Delta_1^wC_\psi(2C_\phi + \alpha_0)$ lead to the desired result. $\qquad\square$

Now, we restate and prove Theorem 4.2.

**Theorem A.9** (Restatement of Theorem 4.2). *Suppose Assumptions 2.1, 2.2, and 2.3 hold. Let $\{x_k, \lambda_k\}_{k\geq 0}$ be the sequence generated by Algorithm 1 such that for any $k \geq 0$, $\alpha_k = \frac{T^{1/3}}{(k+2)^{1+\omega}}$, $\gamma_k = \gamma = \min\{\frac{\mu C_g^{2-2\theta}}{T^{1/3}}, (L_f + L_{xy}^\phi)^{-1}\}$, $N_k = \frac{2}{1-\delta_y}\log(k+1)$, and $M_k = \frac{1}{1-\delta_w}\max\{\max\{1, \frac{1}{2\theta}\}\log(T), \log(T[\psi(x_k, w_k)]_+^{4\theta-2})\}$ if $[\psi(x_k, w_k)]_+\|\nabla_x\psi(x_k, w_k)\| > 0$, otherwise, $M_k = \frac{1}{1-\delta_w}\max\{1, \frac{1}{2\theta}\}\log(T)$. Then, for any $\epsilon > 0$, there exists $t \in \{0, \ldots, T - 1\}$ such that*

    *1. (Stationarity) $\|\nabla f(x_t) + \lambda_t\nabla g(x_t)\| \leq \epsilon$ within $T = \mathcal{O}(\frac{1}{\epsilon^3})$ iterations;*

2. *(Feasibility)* $[g(x_t)]_+ \leq \epsilon$ *within* $T = \mathcal{O}(\frac{1}{\epsilon^{6\theta}})$ *iterations;*

3. *(Slackness)* $|\lambda_t g(x_t)| \leq \epsilon$ *within* $T = \mathcal{O}(\frac{1}{\epsilon^{3\theta/(1-\theta)}})$ *iterations.*

*Proof.* Before starting the proof let us define $t \triangleq \text{argmin}_{0 \leq k \leq T-1} \max\{\|\nabla f(x_k) + \lambda_k \nabla g(x_k)\|, [g(x_k)]_+, |\lambda_k g(x_k)|\}$. Moreover, the selection of parameters $\alpha_k$, $\gamma_k$, $N_k$, and $M_k$ implies that the conditions of Lemma A.7 hold, and we can invoke its result within the proof.

**Part 1.** First, we show the result for $\epsilon$-stationary condition. From the definition of $d_k$ and comparing it with $\nabla f(x_k) + \lambda_k \nabla g(x_k)$ we observe that if $\zeta(x_k, w_k) = 0$ then $\lambda_k = 0$ and $\|\nabla f(x_k) + \lambda_k \nabla g(x_k)\|^2 \leq 2\|d_k\|^2 + 2(L_{xy}^\phi \|y_k - y^*(x_k)\|)^2 \leq 2\|d_k\|^2 + 2(L_{xy}^\phi)^2 \Delta_1^y \delta_y^{N_k}$ which by selecting $N_k$ as in the statement of corollary, we obtain $\|\nabla f(x_k) + \lambda_k \nabla g(x_k)\|^2 \leq 2\|d_k\|^2 + 2(L_{xy}^\phi)^2 \Delta_1^y \frac{1}{(k+1)^2}$. If $\zeta(x_k, w_k) > 0$, then

$$
\begin{aligned}
&\|\nabla f(x_k) + \lambda_k \nabla g(x_k)\|^2 \\
&\leq 3\|d_k\|^2 + 3\|\nabla f(x_k) - \nabla_x \phi(x_k, y_k)\|^2 + 3\lambda_k^2 \|\nabla g(x_k) - \nabla_x \psi(x_k, w_k)\|^2 \\
&\leq 3\|d_k\|^2 + 3(L_{xy}^\phi)^2 \Delta_1^y \delta_y^{N_k} + 3\lambda_k^2 (L_{xw}^\psi)^2 \Delta_1^w \delta_w^{M_k} \\
&\leq 3\|d_k\|^2 + 3(L_{xy}^\phi)^2 \Delta_1^y \delta_y^{N_k} + 3\mu^2 C^2 [\psi(x_k, w_k)]_+^{2-4\theta} (L_{xw}^\psi)^2 \Delta_1^w \delta_w^{M_k} \\
&\leq 3\|d_k\|^2 + 3(L_{xy}^\phi)^2 \Delta_1^y \frac{1}{(k+1)^2} + 3\mu^2 C^2 (L_{xw}^\psi)^2 \Delta_1^w \frac{1}{T},
\end{aligned}
\tag{25}
$$

where in the second inequality we used Lipschitz continuity of $\nabla_x \phi$ and $\nabla_x \psi$ as well as the relations in (15) and (16). The third inequality follows from Lemma A.3 and Assumption 2.3 which shows that $\lambda_k \leq C\|\nabla_x \psi(x_k, w_k)\|^{-1} \leq C\mu[\psi(x_k, w_k)]_+^{1-2\theta}$ for some $C > 0$. The last inequality is obtain by plugging the selection of $N_k$ and $M_k$ as in the statement of corollary and noting that $\frac{1}{1-\delta} \geq 1/\log(1/\delta)$ for any $\delta \in (0, 1)$.

On the other hand, from Theorem A.8 part (I), by selecting $\gamma = \mathcal{O}(1/T^{1/3})$ and $\alpha_k = \frac{T^{1/3}}{(k+2)^{1+\omega}}$ and noting that $\frac{1}{T} \sum_{k=0}^{T-1} \alpha_k = \mathcal{O}(1/T^{2/3})$, we conclude that $\frac{1}{T} \sum_{k=0}^{K-1} \|d_k\|^2 \leq \mathcal{O}(1/T^{2/3})$. Therefore, combining the result with (25) we obtain

$$
\|\nabla f(x_t) + \lambda_t \nabla g(x_t)\|^2 \leq \frac{1}{T} \sum_{k=0}^{K-1} \|\nabla f(x_k) + \lambda_k \nabla g(x_k)\|^2 \leq \mathcal{O}\left(\frac{1}{T^{2/3}} + \frac{(L_{xw}^\psi)^2 \Delta_1^w}{T}\right).
$$

By taking the square root of both sides of the above inequality, the result of part 1 follows immediately.

**Part 2.** From Lemma A.7, we observe that there exists $D > 0$ such that $D = \sum_{k=0}^{T-1} \gamma \|d_k\|^2 < +\infty$. Considering the result of Theorem A.8-part (II), selecting $\gamma_k = \gamma = \mathcal{O}(\frac{1}{T^{1/3}})$, and $p(x) \geq 0$, we have that

$$
\begin{aligned}
\frac{1}{A_T} \sum_{k=0}^{T-1} \alpha_k [g(x_k)]_+^{2\theta} &\leq \frac{\mu}{A_T \gamma} p(x_0) + \frac{\mu}{A_T} \sum_{k=0}^{T-1} \left(\frac{\bar{\Delta}}{\gamma} \delta_w^{M_k} + \frac{2\alpha_k}{\mu} (\Delta_2^w)^{2\theta} \delta_w^{2\theta M_k}\right) + \frac{\mu(L_p + 2L_{xw}^\psi)}{2A_T} D \\
&\leq \mathcal{O}\left(\frac{1}{A_T \gamma} + \frac{D}{A_T}\right),
\end{aligned}
$$

where the last inequality follows from plugging in $M_k$ since $\max\{\delta_w^{M_k}, \delta_w^{2\theta M_k}\} = \mathcal{O}(\frac{1}{T})$. Therefore, from the above inequality, noting that $A_T = \Omega(T^{1/3})$, and the definition of $t$ at the beginning of the proof we conclude that $[g(x_t)]_+^{2\theta} \leq \mathcal{O}(\frac{1}{T^{1/3}})$ which completes the proof of part 2.

**Part 3.** Finally, to calculate the complexity of finding $\epsilon$-complementarity slackness, recall the update of $\lambda_k$ in Algorithm 1. Recall that $\zeta(x_k, w_k)[\psi(x_k, w_k)]_+ \|\nabla_x \psi(x_k, w_k)\|$. If $\zeta(x_k, w_k) = 0$, then $\lambda_k = 0$, hence, $\lambda_k g(x_k) = 0$. Suppose $\zeta(x_k, w_k) > 0$, then we observe that $g(x_k) \geq \psi(x_k, w_k) > 0$. Therefore, from Lemma A.3 we have that $0 \leq \lambda_k g(x_k) = \lambda_k [g(x_k)]_+ \leq C[g(x_k)]_+^{2-2\theta} + C\Delta_2^w \delta_w^{M_k} [\psi(x_k, w_k)]_+^{1-2\theta}$. Combining the two scenarios, for any $k \geq 0$, we have that $|\lambda_k g(x_k)| \leq C[g(x_k)]_+^{2-2\theta} + C\Delta_2^w \delta_w^{M_k} [\psi(x_k, w_k)]_+^{1-2\theta}$ for some $C > 0$. Therefore, based on selection of $M_k$, we obtain $|\lambda_t g(x_t)| \leq \mathcal{O}(\frac{1}{T^{(1-\theta)/(3\theta)}} + \frac{1}{T})$ from which the result follows. $\qquad \square$

### A.5 Experiment Details and Additional Plots

**Experiment Details:** In all experiments, we select the regularization parameter $\lambda = 10^{-3}$ and the maximization variables $y, w$ are updated by running $N_k = 2\lceil \log(k+2)\rceil$ and $M_k = 10\lceil \log(k+2)\rceil$ steps of the projected gradient ascent method. The stepsize $\gamma$ is tuned by selecting the best performance among $\{10^{-4}, 2.5 \times 10^{-4}, 5 \times 10^{-4}, 10^{-3}, 5 \times 10^{-3}, 10^{-2}\}$ and the parameter is set $\alpha_k = \alpha/(k+2)^{1.001}$ for $\alpha \in \{0.1, 0.2, 0.5, 1\}$. Hyperparameter choices follow Theorem 4.2 and tuned via targeted grid search to ensure robustness. Furthermore, to determine the threshold value $r$, we solve the robust learning task in the constraint, i.e, $\min_x \max_{w \in \Delta_m} \sum_{j=1}^{m} \ell_2(x, \xi_j^{(2)}) - g_m(w)$, separately using the unconstrained variant of our method for a some iterations. The resulting objective value is then used in the original problem as the threshold value.

The oscillations that occur in plots reflect the difficult trade-off between minimizing the objective, enforcing feasibility under infinitely many functional constraints, and satisfying the $\epsilon-$KKT conditions, a behavior common in both convex and nonconvex problems with functional constraints [27].

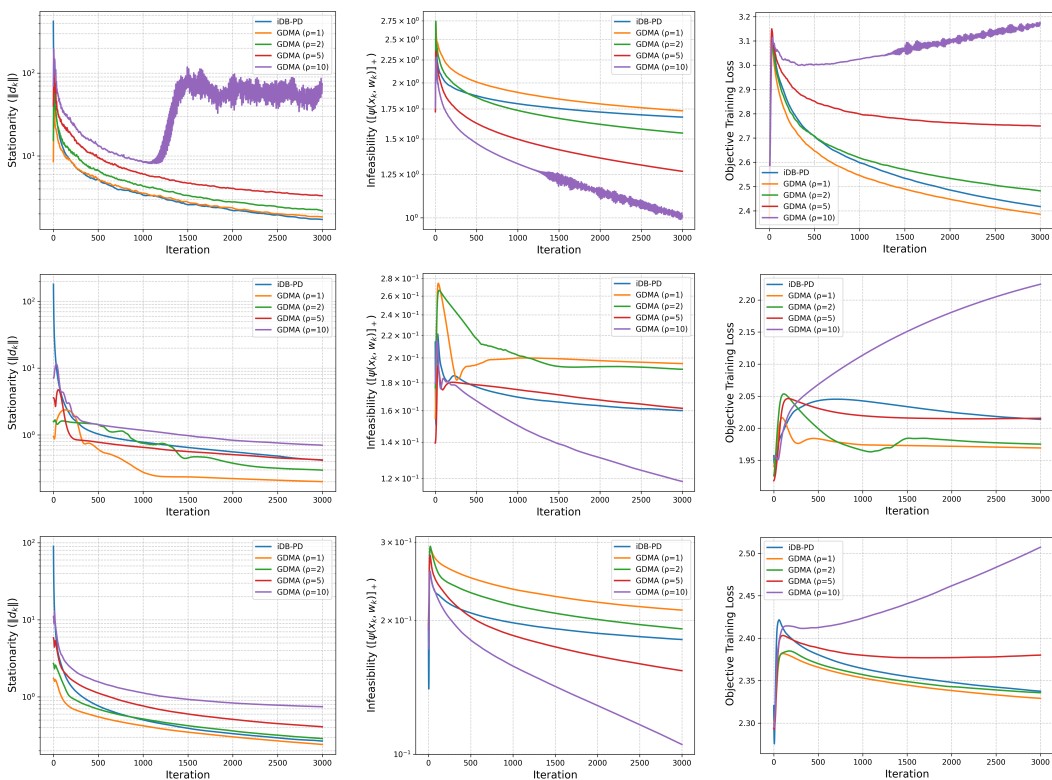

Figure 3: iDB-PD vs. GDMA on multi-Fashion MNIST (top row), Yeast (middle row), and 20NG (bottom row), evaluated in terms of stationarity, infeasibility, and objective loss.

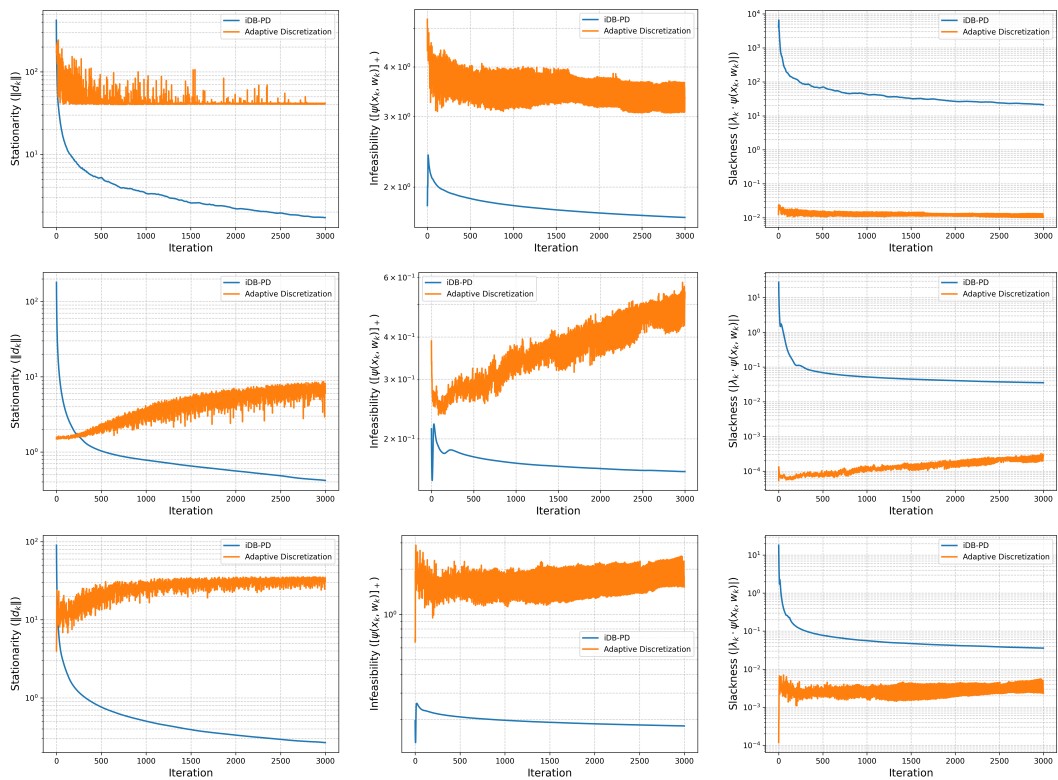

Figure 4: iDB–PD vs. Adaptive Discretization with COOPER on multi-Fashion MNIST (top row), Yeast (middle row), and 20NG (bottom row), evaluated in terms of stationarity, infeasibility, and slackness.

Across the three additional datasets, iDB-PD broadly outperforms all GDMA variants and the adaptive discretization method with COOPER. iDB-PD drives infeasibility and stationarity down quickly while maintaining competitive objective values. In contrast, GDMA requires large penalty values to approach feasibility, frequently at the cost of stability. Further, adaptive discretization struggles with instability and struggles with matching iDB-PD's stationarity and infeasibility performance. These results confirm the robustness of our iDB-PD method which effectively balances feasibility, optimality, and stability.

