# OpenReview forum: "Semi-infinite Nonconvex Constrained Min-Max Optimization"
_NeurIPS.cc/2025/Conference — NeurIPS 2025 poster_

### Official Review · Reviewer_2zSA · 2025-06-23

**Clarity:** 3
**Significance:** 2
**Originality:** 3
**Rating:** 4
**Confidence:** 3

**Summary:**

This paper proposes the iDB-PD method, which adaptively adjusts the search direction based on the satisfaction of the constraint functions. This method is designed for a class of nonconvex minimax optimization problems with nonconvex infinite constraints. This paper further establishes theoretical convergence guarantees under certain assumptions, such as smoothness, strong concavity, or the Polyak–Łojasiewicz (PL) condition.

**Questions:**

1. To approximate the value of $\zeta(x_k, w^{*}(x_k))$, this work uses $\zeta(x_k, w_k)$ where $w_k$ is attained by multi-step gradient ascent and this seems work well under the assumption $\eta_{\psi}$-strongly concavity or $c_{\psi}$-PL condition. However, I believe the number of steps for the multi-step gradient ascent should depend on constants like $\eta_{\psi}$ or $c_{\psi}$ to guarantee the attained value is well-approximated. Additionally, the required number of steps seems related to the value of $\theta$. Since these problem-dependent constants are not given a priori, how can the exact number of steps, $M_k$, be practically selected?
2. The performance in the Multi-MNIST experiment described in Figure 1 exhibits significant oscillation. It would be helpful if the authors could provide some explanation for this behavior.
3. (Minor) There appears to be a typo on line 240; it should likely be "whether."

**Ethical Concerns:**

["NO or VERY MINOR ethics concerns only"]

**Final Justification:**

The author has provided satisfactory answers to my questions and concerns regarding both the practical selection of hyper-parameters and explanations of the given experiment. Therefore, I will maintain my positive score.

**Limitations:**

yes

**Quality:**

3

**Strengths And Weaknesses:**

Strengths:
1. This work establishes global non-asymptotic convergence guarantees under a more generalized regularity condition.
2. This paper is well organized and effectively presents the key differences, making it easy for the reader to follow.

Weaknesses: While the theoretical results are well-developed, it is not entirely clear how they can be effectively applied in practice; see the following question for more details.

---

> ### Author Rebuttal · Authors · 2025-07-30
>
> **1. The number of steps for the multi-step gradient ascent should depend on constants like $\eta_\psi$ or $c_\psi$. Additionally, the required number of steps seems related to the value of $\theta$. How can the exact number of steps, $M_k$, be practically selected?**
>
> **R1.**
> We thank the reviewer for their comment. The multi-step parameters $M_k$ and $N_k$ indeed depend on problem-specific constants associated with the maximization component, such as $\eta_\psi$ and $c_\psi$. Details regarding the selection of these parameters are provided in Appendix Theorem A.9 ( Restatement of Theorem 4.2). Additionally, the dependence of $M_k$ and $N_k$ on the parameter $\theta$ has an intuitive interpretation: $\theta$ reflects the complexity of the infinite constraint set, where values of $\theta$ closer to 0 indicate easier subproblems and faster convergence, while values near 1 correspond to more challenging scenarios and slower convergence rates -- See Remark 4.1.
>
> In practice, although the exact problem constants may be unknown, the theoretical results in Theorem 4.2 suggest that $M_k$ and $N_k$ should grow at a rate of at least $\mathcal{O}(\log(k+2))$ or $\mathcal{O}(\log(T+1))$, where $T$ denotes the total number of iterations. One practical strategy is to choose a slightly more aggressive growth, e.g., $\lceil (k+2)^{0.1}\rceil$ or $\lceil \log^2(k+2)\rceil$, ensuring that beyond some iteration index $T_0$, their values exceed the theoretical threshold. As a result, the convergence guarantees of the proposed method remain valid after a certain warm-up phase.
>
>
> ---
>
>
> **2. The performance in the Multi-MNIST experiment described in Figure 1 exhibits significant oscillation. It would be helpful if the authors could provide some explanation for this behavior.**
>
> **R2.** This behavior indeed stems from the inherent complexity of the problem. As noted in lines 359–361, our robust Multi-Task Learning formulation gives rise to a challenging nonconvex optimization problem with infinitely many functional constraints. Although our method is supported by provable convergence guarantees, the observed oscillations are a natural manifestation of the algorithm's iterative efforts to balance three competing goals: minimizing the objective function, enforcing feasibility with respect to the robust constraints, and satisfying the KKT optimality conditions (including stationarity and complementarity slackness). These oscillations reflect the algorithm's trajectory as it alternates between improving feasibility and achieving stationarity. This behavior commonly arises in both convex and nonconvex optimization problems with functional constraints, e.g., see the experiments in [Ref1].
>
> [Ref1] Yazdandoost Hamedani, Erfan, and Necdet Serhat Aybat. "A primal-dual algorithm with line search for general convex-concave saddle point problems." SIAM Journal on Optimization 31.2 (2021): 1299-1329.
>
> ---
>
>
> **3. (Minor) There appears to be a typo on line 240; it should likely be "whether."**
>
> **R3.**
> We thank the reviewer for pointing out this typo. All typos will be corrected in the revised version of the manuscript.
>
> ---
>
> We hope our responses address your concerns and would be happy to clarify anything further if needed.

---

> > ### Comment · Reviewer_2zSA · 2025-08-04
> >
> > Thank you for the thorough response. As my main concerns have been resolved, I will keep my positive score.

---

### Official Review · Reviewer_E7VL · 2025-07-01

**Clarity:** 2
**Significance:** 2
**Originality:** 3
**Rating:** 3
**Confidence:** 3

**Summary:**

This paper studies semi-infinite constrained min-max optimization problems and proposes a novel method that does not rely on the commonly assumed convexity of the objective function and constraint functions, as required by many prior works.
The author(s) also prove that their algorithm achieves an $\epsilon$-KKT solution within a finite number of iterations.
Finally, they demonstrate the effectiveness of their approach by applying it to a multi-task learning problem.

**Questions:**

Please see weaknesses

**Ethical Concerns:**

["NO or VERY MINOR ethics concerns only"]

**Final Justification:**

The direction is new and worth exploring, but on the other hand, it's a bit unclear to me yet if the proposed approach will eventually be widely applied.

**Quality:**

2

**Strengths And Weaknesses:**

#### Strengths
- The author(s) study semi-infinite nonconvex constrained min-max optimization problems and propose a solution approach that does not require the commonly assumed convexity of the objective or constraint functions.
- The author(s) provides detailed proofs.

#### Weaknesses

- Since the proposed method is designed to handle both convex and non-convex functions, it is unclear how it compares to discretization or cutting surface approaches in the convex setting. On the other hand, the authors stated that "most classical SIP metnods assume convexity or rely on simplifications such as discrete approximation ..." So even for non-convex cases, simple discretization should be possible. Shouldn't it be used as a baseline in the experiments for comparing both performance and efficiency? Currently, the authors did not compare with any methods.

- The experimental section formulates multi-task learning as a non-convex DRO problem and applies the proposed method to solve it.
While this is an interesting direction, it is not evident whether this setup provides clear advantages over more standard approaches, such as minimizing the sum of task-specific losses.

- To achieve $\epsilon$-KKT conditions, the proof heavily relies on the appropriate selection of hyperparameters such as $\alpha_k$, $\gamma_k$, $N_k$, and $M_k$.
However, in the experimental section, the author(s) did not verify whether the chosen values for these hyperparameters satisfy the theoretical assumptions.

- In Section 3, the author(s) propose the indicator function $\zeta(x, w)$ and claim that if this function is zero, then the point $x$ is feasible (i.e., the residual is zero) and a critical point of the constraint function (i.e., the gradient value is zero).
However, this claim is incorrect, as $\zeta(x, w^*(x)) = 0$ only guarantees that at least one of the two terms in the indicator function is zero, but not necessarily both.

---

> ### Author Rebuttal · Authors · 2025-07-30
>
> **1. How it compares to discretization or cutting surface approaches in the convex setting? Even for non-convex cases, simple discretization should be possible. Shouldn't it be used as a baseline in the experiments for comparing both performance and efficiency?**
>
> **R1.** We appreciate the reviewer's comment. Classical SIP methods focus on minimization problems, and we are not aware of any method that can address min-max structure in the objective function in a nonconvex setting. Moreover, those that can handle nonconvex problems generally focus on a discretization technique, however, these methods suffer from the curse of dimensionality by construction, as the number of required grid points to accurately represent the infinite constraint naturally grows with the dimension of the constraint space and is not effectively implementable on our numerical examples involving neural networks in high dimension (dim($x$)$\approx 2M$). On the other hand, our proposed method offers a clear advantage by relying solely on first-order information. Other common SIP methods, such as cutting planes or exchange methods, rely on a convexity assumption and do not offer convergence guarantees in the non-convex setting.
> Common constrained min-max methods, such as gradient-based primal-dual methods require the constraint set to be tractable via projection or evaluation. However, in our semi-infinite constraint setting, there are no guarantees that either will hold.
>
> For the baseline comparison, we consider an additive robust MTL model where the objective function is the weighted summation of task-specific losses. More specifically, we consider solving $\min_{x}\max_{y\in \Delta_n}\max_{w\in\Delta_m}\phi(x,y)+\rho\psi(x,w)$ where $\rho>0$ is the weight parameter. This problem can be solved via min-max optimization methods such as gradient descent multi-step ascent (GDMA) [Ref 1,Ref 2]. We compare the performance of our method GDMA for different values of $\rho$ in terms of the first task loss, i.e, $\phi(x_k,y_k)$, and feasibility $[\psi(x_k,w_k)-r]_+$, where $r$ is the single task threshold as discussed in our paper. Below you can find the result of the experiment on Multi-MNIST dataset.
>
> **iDB-PD (our method)**
>
> iteration = 0001: Obj Loss = 2.4279 Feasibility = 1.0807
>
> iteration = 1000: Obj Loss = 2.8115 Feasibility = 0.2393
>
> iteration = 2000: Obj Loss = 2.7242 Feasibility = 0.0387
>
> iteration = 3000: Obj Loss = **2.5080** Feasibility = **0.0010**
>
> **GDMA ($\rho=1$)**
>
> iteration = 0001: Obj Loss = 2.4073 Feasibility = 1.0576
>
> iteration = 1000: Obj Loss = 2.2709 Feasibility = 0.9157
>
> iteration = 2000: Obj Loss = 2.1412 Feasibility = 0.7698
>
> iteration = 3000: Obj Loss = 2.0615 Feasibility = 0.6823
>
> **GDMA ($\rho=2$)**
>
> iteration = 0001: Obj Loss = 2.4556 Feasibility = 1.1168
>
> iteration = 1000: Obj Loss = 2.3824 Feasibility = 0.6930
>
> iteration = 2000: Obj Loss = 2.2876 Feasibility = 0.5165
>
> iteration = 3000: Obj Loss = 2.2276 Feasibility = 0.3941
>
> **GDMA ($\rho=5$)**
>
> iteration = 0001: Obj Loss = 2.4700 Feasibility = 1.0810
>
> iteration = 1000: Obj Loss = 2.6608 Feasibility = 0.3444
>
> iteration = 2000: Obj Loss = 2.6859 Feasibility = 0.0352
>
> iteration = 3000: Obj Loss = 2.7191 Feasibility = 0.0000
>
> **GDMA ($\rho=10$)**
>
> iteration = 0001: Obj Loss = 2.4803 Feasibility = 1.0565
>
> iteration = 1000: Obj Loss = 3.1297 Feasibility = 0.0000
>
> iteration = 2000: Obj Loss = 3.4119 Feasibility = 0.0000
>
> iteration = 3000: Obj Loss = 3.5966 Feasibility = 0.0000
>
>
> We observe that our proposed approach reaches the lowest value for the first task loss among the approaches that satisfy the constraint within an accuracy $\epsilon=10^{-3}$. Note that while GDMA with $\rho=1$ yields a lower objective loss, it fails to meet the feasibility requirement within the desired accuracy threshold. The full details and corresponding plots for the experiment will be added to the revised manuscript.
>
> [Ref 1] Jin, Chi, Praneeth Netrapalli, and Michael Jordan. "What is local optimality in nonconvex-nonconcave minimax optimization?." International conference on machine learning. PMLR, 2020.
>
> [Ref 2] Nouiehed, Maher, et al. "Solving a class of non-convex min-max games using iterative first order methods." Advances in Neural Information Processing Systems 32 (2019).
>
>
> ---
>
>
> **2. It is not evident whether this setup provides clear advantages over more standard approaches, such as minimizing the sum of task-specific losses.**
>
> **R2.** We appreciate the reviewer’s comment. In our revised experiments, we include a comparison with the robust formulation that minimizes the sum of task-specific losses. As detailed in our response to Question 1, our proposed method, iDB-PD, consistently achieves lower loss values across both tasks compared to GDMA with $\rho = 1$, highlighting the effectiveness of our formulation.
>
> Furthermore, recent studies [Ref 3, Ref 4] have demonstrated the benefits of explicitly modeling task priorities in multi-task learning. Minimizing the unweighted sum of task-specific losses inherently assumes equal importance for all tasks, which limits the model’s ability to adapt to task-specific needs or real-world constraints where some tasks may be more critical. In contrast, our robust MTL formulation enables more flexible and effective optimization across heterogeneous tasks.
>
> [Ref 3] Cheng, Zhengxing, et al. "No More Tuning: Prioritized Multi-Task Learning with Lagrangian Differential Multiplier Methods." Proceedings of the AAAI Conference on Artificial Intelligence. Vol. 39. No. 11. 2025.
>
> [Ref 4] Guo, Michelle, et al. "Dynamic task prioritization for multitask learning." Proceedings of the European conference on computer vision (ECCV). 2018.
>
>
> ---
>
>
> **3. In the experimental section, the author(s) did not verify whether the chosen values for these hyperparameters satisfy the theoretical assumptions.**
>
> **R3.** Details of our experimental setup and hyperparameter selection are provided in Appendix A.5. We selected the hyperparameters based on Theorem 4.2 and performed a grid search over a carefully chosen range of values, which is a common approach in the optimization literature, ensuring both theoretical consistency and empirical robustness.
>
> ---
>
>
> **4. In Section 3, $\zeta(x,w^*(x))$ only guarantees that at least one of the two terms in the indicator function is zero, but not necessarily both.**
>
> **R4.** We thank the reviewer for pointing this out. The correct statement should be: *“then the point $x$ is feasible (i.e., the residual is zero) **or** a critical point of the constraint function.”* However, under Assumption 2.3 and using the definition $\zeta(x, w) = ||\nabla_x \psi(x, w) [\psi(x, w)]_+||$, we can indeed conclude that $\zeta(x, w^*(x)) = 0$ implies that $x$ is feasible. We will clarify this point in the revised manuscript by expanding the explanation and explicitly connecting it to Assumption 2.3.
>
> ---
>
> We hope our responses address your concerns and would be happy to clarify anything further if needed.

---

> > ### Comment · Reviewer_E7VL · 2025-08-04
> >
> > Thank the authors for answering my technical questions. On the one hand, the setting is new and it is worth exploring the direction, but on the other hand, it's a bit unclear to me yet if the proposed approach will eventually be widely applied.

---

> > > ### Author Response · Authors · 2025-08-04
> > >
> > > Thank you for your thoughtful feedback. We would like to highlight that semi-infinite programming with min-max objective naturally arises in various applications, including robust learning, robotics, and reinforcement learning with safety constraints--see the citations within the paper and [Ref 1] for a comprehensive review. Currently, applied researchers often simplify/reformulate the problem when facing such complicated formulations or custom-design heuristic algorithms due to the lack of efficient methods with provable guarantees. Our work aims to bridge this gap and initiate a line of research into first-order methods for nonconvex semi-infinite min-max problems, thereby broadening their application. To support broader impact and adoption, we will make our algorithm implementation publicly available through our GitHub website as well.
> > >
> > > [Ref 1] Moos, Janosch, et al. "Robust reinforcement learning: A review of foundations and recent advances." Machine Learning and Knowledge Extraction 4.1 (2022): 276-315.

---

### Official Review · Reviewer_DGYP · 2025-07-02

**Clarity:** 3
**Significance:** 3
**Originality:** 3
**Rating:** 4
**Confidence:** 4

**Summary:**

This paper studies a challenging class of min-max optimization problems with infinitely many nonconvex constraints, motivated by applications such as adversarial robustness and safety-constrained learning, as presented in eq. (1). It considers the setting where the min-max objective $\phi(x,y)$ is Lipschitz in x, and PL or strongly-concave (SC) in y,  and the inequality function $\varphi(x,y)$ is  $\theta$-Lojasiewicz and Lipschitz in x and PL or SC in w. The authors propose an Inexact Dynamic Barrier Primal-Dual (iDB-PD) algorithm and establish non-asymptotic convergence guarantees under the Łojasiewicz inequality. Theoretical results are supported by experiments on robust multitask learning with task prioritization.

**Questions:**

1. In the first paragraph of the introduction, several applications are mentioned without citations. Could references or concrete examples be added?

2. In lines 36–37, the problem structure is emphasized before motivating applications are introduced. Could a simplified application be included earlier?

3. Line 42 introduces nonconvexity in $\phi$ and $\psi$, yet convergence relies on the Łojasiewicz inequality. Why is this assumption not highlighted sooner?

4. Assumption 2.2 sets $W = \mathbb{R}^l$, but in Equation (1), $w \in W$. Please clarify the domain and generality of $W$.

5. Do the applications in Section 1.2 and experiments rigorously satisfy the PL or Łojasiewicz conditions?

6. How does the algorithm's complexity compare to existing SIP or constrained min-max solvers, even heuristically?

7. Is it possible to include baselines or ablation studies to better evaluate the practical performance of the method?

**Ethical Concerns:**

["NO or VERY MINOR ethics concerns only"]

**Final Justification:**

I will keep my positive score.

**Limitations:**

Yes

**Quality:**

3

**Strengths And Weaknesses:**

Strength:

1. The paper addresses a novel and underexplored intersection between semi-infinite programming (SIP) and nonconvex min-max optimization.

2. It provides the first non-asymptotic complexity guarantees for this class of problems, representing a significant theoretical advancement.

3. The algorithm is thoughtfully designed, notably avoiding the common but restrictive assumption of dual boundedness.

4. The theoretical results are rigorous, well-structured, and clearly presented.

5. The application to robust multitask learning is both relevant and practically meaningful.

Weaknesses:

1. The introduction lacks citations to support its claims regarding real-world applications.

2. Motivating applications are introduced too late, which makes the early sections abstract and harder to engage with.

3. The Łojasiewicz inequality assumption plays a critical role but is not adequately emphasized or discussed in the main text.

4. There is inconsistency in the definition of the constraint domain $W$, particularly between Equation (1) and Assumption 2.2.3.

5. The strong assumptions, such as PL and Łojasiewicz conditions, are not verified or discussed in the context of the examples or existing literature.

6. The paper lacks experimental comparisons with baseline or approximate methods, leaving questions about scalability and practical computational cost unanswered.

---

> ### Author Rebuttal · Authors · 2025-07-30
>
> **1. In the first paragraph of the introduction, several applications are mentioned without citations. Could references or concrete examples be added?**
>
> **R1.**
> We thank the reviewer for this helpful suggestion. We agree that the applications mentioned in the opening paragraph would benefit from concrete citations to better support the context and motivate the relevance of semi-infinite programming (SIP).
> Accordingly, we have revised the paragraph to explicitly cite representative works for each application area.
> Specifically, we now reference foundational works highlighting adversarial perturbations in AI models [Ref 1], [Ref 2], robust optimization in supply chain management [Ref 3], and safety and control under uncertainty in autonomous systems [Ref 4].
>
>
> [Ref 1] Goodfellow, Ian J., Jonathon Shlens, and Christian Szegedy. "Explaining and harnessing adversarial examples." arXiv preprint arXiv:1412.6572 (2014).
>
> [Ref 2] Zhang, Xingwei, Xiaolong Zheng, and Wenji Mao. "Adversarial perturbation defense on deep neural networks." ACM Computing Surveys (CSUR) 54.8 (2021): 1-36.
>
> [Ref 3] Ben-Tal, Aharon, et al. "Retailer-supplier flexible commitments contracts: A robust optimization approach." Manufacturing and Service Operations Management 7.3 (2005): 248-271.
>
> [Ref 4] Dutta, Raj Gautam, Xiaolong Guo, and Yier Jin. "Quantifying trust in autonomous system under uncertainties." 2016 29th IEEE International System-on-Chip Conference (SOCC). IEEE, 2016.
>
>  ---
>
>  **2. In lines 36–37, the problem structure is emphasized before motivating applications are introduced. Could a simplified application be included earlier?**
>
> **R2.** Thank you for the suggestion. We agree that introducing a motivating application earlier can improve readability. We will revise the manuscript to include a brief mention of relevant applications with appropriate references, before presenting the problem structure.
>
>  ---
>
>  **3. Line 42 introduces nonconvexity in $\phi$ and $\psi$, yet convergence relies on the Łojasiewicz inequality. Why is this assumption not highlighted sooner.**
>
> **R3.** Thank you for this observation. To clarify, Line 42 only motivates the development of new methods for addressing nonconvexity in both objective and constraint functions under min-max setting as a general direction. Moreover, Łojasiewicz condition is only required for the $\psi$ function with respect to $x$, which we explicitly discussed in the contribution section. We will revise the earlier sections to make this distinction more explicit and avoid potential confusion.
>
> ---
>
> **4. Assumption 2.2 sets $W\in \mathbb R^\ell$, but in Equation (1), $w\in W$. Please clarify the domain and generality of $W$.**
>
> **R4.** Thank you for pointing this out. We agree that the notation could be made clearer.
>
> * When $\psi(x, \cdot)$ is $\eta_\psi$-strongly concave, $W$ can be any closed convex subset of $\mathbb{R}^\ell$.
> * When $\psi(x, \cdot)$ is assumed to satisfy the Polyak-Łojasiewicz (PL) condition with constant $c_\psi$, we require that $W = \mathbb{R}^\ell$ to ensure the PL condition holds over the entire space.
>
> We will revise the assumption to make this distinction more precise in the revised version.
>
> ---
>
> **5. Do the applications in Section 1.2 and experiments rigorously satisfy the PL or Łojasiewicz conditions?**
>
> **R5.** We thank the reviewer for this question. Both applications in Section 1.2 satisfy the Łojasiewicz inequality if the loss functions chosen are real analytical, definable in $o$-minimal structures, or differentiable sub-analytical functions. Common constraint loss choices in both problems include cross-entropy, squared loss, $\ell_1$, and Huber loss, all of which satisfy one of the above functional categories and therefore, the Łojasiewicz inequality as well. Our neural network in all of our experiments is connected by hyperbolic tangents with a softmax activation function, so the network satisfies the Łojasiewicz inequality, as we note in the discussion after Definition 2.1.
>
> ---
>
> **6. How does the algorithm's complexity compare to existing SIP or constrained min-max solvers, even heuristically?**
>
> **R6.** We appreciate the reviewer noting this. Classical SIP methods that can handle non-convex problems generally focus on a discretization technique. However, these methods suffer from the curse of dimensionality by construction, as the number of required grid points to accurately represent the infinite constraint naturally grows with the dimension of the constraint space. Given our numerical examples involve neural networks in high dimension (dim($x$)$\approx 2M$), our proposed method offers a clear advantage by relying solely on first-order information. Other common SIP methods, such as cutting planes or exchange methods, rely on a convexity assumption and do not offer convergence guarantees in the non-convex setting.
> Common constrained min-max methods, such as gradient-based primal-dual methods require the constraint set to be tractable via projection or evaluation. However, in our semi-infinite constraint setting, there are no guarantees that either will hold.
>
> ---
>
> **7. Is it possible to include baselines or ablation studies to better evaluate the practical performance of the method?**
>
> **R7.** We thank the reviewer for their suggestion. The main reason that we originally did not use any baseline method was that classical SIP methods mainly focus on minimization problems, and we are not aware of any method that can address a semi-infinite min-max structure in a nonconvex setting. For the baseline comparison, we consider an additive robust MTL model where the objective function is the weighted summation of task-specific losses. More specifically, we consider solving $\min_{x}\max_{y\in \Delta_n}\max_{w\in\Delta_m}\phi(x,y)+\rho\psi(x,w)$ where $\rho>0$ is the weight parameter. This problem can be solved via min-max optimization methods such as gradient descent multi-step ascent (GDMA) [Ref 5,Ref 6]. We compare the performance of our method GDMA for different values of $\rho$ in terms of the first task loss, i.e, $\phi(x_k,y_k)$, and feasibility $[\psi(x_k,w_k)-r]_+$, where $r$ is the single task threshold as discussed in our paper. Below you can find the result of the experiment on Multi-MNIST dataset.
>
> **iDB-PD (our method)**
>
> iteration = 0001: Obj Loss = 2.4279 Feasibility = 1.0807
>
> iteration = 1000: Obj Loss = 2.8115 Feasibility = 0.2393
>
> iteration = 2000: Obj Loss = 2.7242 Feasibility = 0.0387
>
> iteration = 3000: Obj Loss = **2.5080** Feasibility = **0.0010**
>
> **GDMA ($\rho=1$)**
>
> iteration = 0001: Obj Loss = 2.4073 Feasibility = 1.0576
>
> iteration = 1000: Obj Loss = 2.2709 Feasibility = 0.9157
>
> iteration = 2000: Obj Loss = 2.1412 Feasibility = 0.7698
>
> iteration = 3000: Obj Loss = 2.0615 Feasibility = 0.6823
>
> **GDMA ($\rho=2$)**
>
> iteration = 0001: Obj Loss = 2.4556 Feasibility = 1.1168
>
> iteration = 1000: Obj Loss = 2.3824 Feasibility = 0.6930
>
> iteration = 2000: Obj Loss = 2.2876 Feasibility = 0.5165
>
> iteration = 3000: Obj Loss = 2.2276 Feasibility = 0.3941
>
> **GDMA ($\rho=5$)**
>
> iteration = 0001: Obj Loss = 2.4700 Feasibility = 1.0810
>
> iteration = 1000: Obj Loss = 2.6608 Feasibility = 0.3444
>
> iteration = 2000: Obj Loss = 2.6859 Feasibility = 0.0352
>
> iteration = 3000: Obj Loss = 2.7191 Feasibility = 0.0000
>
> **GDMA ($\rho=10$)**
>
> iteration = 0001: Obj Loss = 2.4803 Feasibility = 1.0565
>
> iteration = 1000: Obj Loss = 3.1297 Feasibility = 0.0000
>
> iteration = 2000: Obj Loss = 3.4119 Feasibility = 0.0000
>
> iteration = 3000: Obj Loss = 3.5966 Feasibility = 0.0000
>
>
> We observe that our proposed approach reaches the lowest value for the first task loss among the approaches that satisfy the constraint within an accuracy $\epsilon=10^{-3}$. Note that while GDMA with $\rho=1$ yields a lower objective loss, it fails to meet the feasibility requirement within the desired accuracy threshold. The full details and corresponding plots for the experiment will be added to the revised manuscript.
>
> [Ref 5] Jin, Chi, Praneeth Netrapalli, and Michael Jordan. "What is local optimality in nonconvex-nonconcave minimax optimization?." International conference on machine learning. PMLR, 2020.
>
> [Ref 6] Nouiehed, Maher, et al. "Solving a class of non-convex min-max games using iterative first order methods." Advances in Neural Information Processing Systems 32 (2019).
>
>
> ---
>
> We hope our responses address your concerns and would be happy to clarify anything further if needed.

---

> > ### Comment · Reviewer_DGYP · 2025-08-03
> >
> > Thank you. My concerns are mostly resolved.

---

### Official Review · Reviewer_hjym · 2025-07-03

**Clarity:** 3
**Significance:** 2
**Originality:** 2
**Rating:** 5
**Confidence:** 4

**Summary:**

The paper develops an optimization algorithm for a min-max problem with a nonconvex objective that depends jointly on the minimization and maximization variables, while the minimization variables must also satisfy a semi-infinite family of constraints. The authors introduce a first-order dynamic barrier primal-dual method that behaves like a single-loop primal-dual gradient scheme augmented with a barrier term, and prove strong convergence rates under a uniform Łojasiewicz property, required since there is no convexity imposed. The authors further demonstrate that distributionally robust multi-task learning and related applications can be formulated within this framework and efficiently solved by the proposed algorithm.

**Questions:**

- Would Assumption 2.3 not hold for NNs in general?
- Do the authors believe simplex constraints are representative enough for $\psi$ in the numerical experiments?

**Ethical Concerns:**

["NO or VERY MINOR ethics concerns only"]

**Final Justification:**

The literature review still feels underdeveloped. I do not know why the authors cannot have any benchmarks, since there are so many SIO problems in RO. The rest looks good.

Note: the authors' second reply is much more thorough. I increased my score to 5. No more confusion on my end.

**Limitations:**

Yes.

**Paper Formatting Concerns:**

No concern about the formatting of the paper.

**Quality:**

3

**Strengths And Weaknesses:**

The paper is written very well. I was able to follow most of the results presented. I believe the subject is of interest to our community, and I thank the authors for this work. The convergence guarantees look appealing.

I have some concerns that I am listing below. I would like to discuss the following points with the authors during the rebuttal phase.

- Firstly, a lot of times, the paper specifies that semi-infinite optimization is underdeveloped. I disagree with this statement. SIO is one of the most well-studied concepts in robust optimization and related fields. A lot of important citations and discussions are missing in this work. For example, the book by Anderson and Nash ("Linear Programming in Infinite-dimensional Spaces") is missing. The citations in robust optimization are also not usual in my view, especially since Ben-Tal's book is missing (which has a lot of useful SIO results).
- There are a lot of works in distributionally robust optimization (*e.g.*, "regularization via mass transportation") where such min-max problems look nonconvex but the inner problem can be dualized and we can still tractably solve the problem. In those cases, the inner "convex maximization" problem can still be dualized with some modeling tricks. Such approaches are not really discussed in this work, and all the existing results are discarded by using "nonconvexity" as an argument.
- The approach proposed in this work can actually be useful, since I agree that the assumptions are very mild. However, numerical experiments have no benchmark to compare against. The paper originally claimed that this is a very important class of problems, but I find it difficult to be convinced by the importance of the class of problems if numerical experiments do not have any benchmarks. In my view, to motivate the proposed sophisticated algorithms, we should get strong numerical evidence. Some benchmark thoughts: we can solve one task in the multi-task learning setting and test the performance of our trained models on the other task (an extreme approach). We can use local optimization solvers with multiple starts simulated. We can use bilevel optimization solvers. We can try to linearized the objective function (e.g., approximate $\psi$ with piece-wise affine functions).

---

> ### Author Rebuttal · Authors · 2025-07-30
>
> **1. The paper specifies that semi-infinite optimization is underdeveloped. I disagree with this statement. A lot of important citations and discussions are missing in this work. The citations in robust optimization are also not usual in my view, especially since Ben-Tal's book is missing.**
>
> **R1.**
> We would like to thank the reviewer for their thoughtful and constructive comment. Our intention was certainly not to overlook or diminish the extensive theoretical and algorithmic contributions in the SIP literature. Rather, we aimed to emphasize that, despite its rich development and broad applications in various areas such as robotic and control theory, comparatively less attention has been given to methods tailored for modern AI and Operations Research problems, which often involve different structural features (e.g., min-max objective functions) and require different computational approaches (e.g., fully first-order methods). To clarify this point, we will revise our statement as follows: *"Despite its rich and extensive theoretical and algorithmic development, comparatively less attention has been devoted to designing efficient first-order methods for emerging applications in modern AI and Operations Research."* Additionally, we expand our literature review to include more classical works in SIP, including those kindly suggested by the reviewer.
>
> Our work introduces **the first fully first-order algorithm** for **nonconvex semi-infinite constrained min-max optimization**, that provides **explicit global non-asymptotic convergence guarantees** for finding an approximated stationary solution. Unlike prior SIP methods which are often second-order, require feasible initialization, and offer only asymptotic or local convergence results, our iDB-PD algorithm ensures convergence from any starting point, with provable iteration complexity bounds of $\mathcal{O}(\epsilon^{-3})$, $\mathcal{O}(\epsilon^{-6\theta})$, and $\mathcal{O}(\epsilon^{-3\theta/(1-\theta)})$ for stationarity, feasibility, and complementarity, respectively. In contrast, classical methods like exchange algorithms or adaptive discretization (e.g., [Ref 1], [Ref 2]) either lack convergence rate analysis or provide only local rates (e.g., quadratic convergence near optimality under strong assumptions), and robust optimization frameworks (e.g., Ben-Tal et al.) solve restricted convex subclasses without iterative rate analysis. Thus, our method uniquely bridges the gap between theoretical rigor and broad applicability, advancing the state of the art in semi-infinite programming.
>
> [Ref 1] Seidel, Tobias, and Karl-Heinz Küfer. "An adaptive discretization method solving semi-infinite optimization problems with quadratic rate of convergence." Optimization 71.8 (2022): 2211-2239.
>
> [Ref 2] Floudas, Christodoulos A., and Oliver Stein. "The adaptive convexification algorithm: a feasible point method for semi-infinite programming." SIAM Journal on Optimization 18.4 (2008): 1187-1208.
>
>
> ---
>
> **2. There are a lot of works in distributionally robust optimization. In those cases, the inner "convex maximization" problem can still be dualized with some modeling tricks. Such approaches are not really discussed in this work.**
>
> **R2.**
> We appreciate the reviewer’s comment. While we acknowledge the extensive literature on distributionally robust optimization (DRO), including approaches such as “regularization via mass transportation” where the inner maximization can be dualized under convexity assumptions, our work addresses a fundamentally different setting: nonconvex-concave semi-infinite min-max problems where such dualization does not lead to a tractable convex reformulation. Unlike prior DRO methods that rely on strong duality and convexity to enable tractable solutions, our goal is to develop a fully first-order method with global non-asymptotic convergence guarantees. We will explicitly incorporate this distinction and relevant DRO references in the revised paper.
>
> ---
>
> **3. Numerical experiments have no benchmark to compare against. In my view, to motivate the proposed sophisticated algorithms, we should get strong numerical evidence.**
>
> **R3.**
> Thank you for your comment and great suggestion. The main reason that we originally did not use any baseline method was that classical SIP methods mainly focus on minimization problems, and we are not aware of any method that can address a semi-infinite min-max structure in a nonconvex setting. Moreover, those that can handle non-convex problems generally focus on a discretization technique, however, these methods suffer from the curse of dimensionality by construction, as the number of required grid points to accurately represent the infinite constraint naturally grows with the dimension of the constraint space and is not effectively implementable on our numerical examples involving neural networks in high dimension (dim($x$)$\approx 2M$).
> For the baseline comparison, we consider an additive robust MTL model where the objective function is the weighted summation of task-specific losses. More specifically, we consider solving $\min_{x}\max_{y\in \Delta_n}\max_{w\in\Delta_m}\phi(x,y)+\rho\psi(x,w)$ where $\rho>0$ is the weight parameter. This problem can be solved via min-max optimization methods such as gradient descent multi-step ascent (GDMA) [Ref 3,Ref 4]. We compare the performance of our method GDMA for different values of $\rho$ in terms of the first task loss, i.e, $\phi(x_k,y_k)$, and feasibility $[\psi(x_k,w_k)-r]_+$, where $r$ is the single task threshold as discussed in our paper. Below you can find the result of the experiment on Multi-MNIST dataset.
>
> **iDB-PD (our method)**
>
> iteration = 0001: Obj Loss = 2.4279 Feasibility = 1.0807
>
> iteration = 1000: Obj Loss = 2.8115 Feasibility = 0.2393
>
> iteration = 2000: Obj Loss = 2.7242 Feasibility = 0.0387
>
> iteration = 3000: Obj Loss = **2.5080** Feasibility = **0.0010**
>
> **GDMA ($\rho=1$)**
>
> iteration = 0001: Obj Loss = 2.4073 Feasibility = 1.0576
>
> iteration = 1000: Obj Loss = 2.2709 Feasibility = 0.9157
>
> iteration = 2000: Obj Loss = 2.1412 Feasibility = 0.7698
>
> iteration = 3000: Obj Loss = 2.0615 Feasibility = 0.6823
>
> **GDMA ($\rho=2$)**
>
> iteration = 0001: Obj Loss = 2.4556 Feasibility = 1.1168
>
> iteration = 1000: Obj Loss = 2.3824 Feasibility = 0.6930
>
> iteration = 2000: Obj Loss = 2.2876 Feasibility = 0.5165
>
> iteration = 3000: Obj Loss = 2.2276 Feasibility = 0.3941
>
> **GDMA ($\rho=5$)**
>
> iteration = 0001: Obj Loss = 2.4700 Feasibility = 1.0810
>
> iteration = 1000: Obj Loss = 2.6608 Feasibility = 0.3444
>
> iteration = 2000: Obj Loss = 2.6859 Feasibility = 0.0352
>
> iteration = 3000: Obj Loss = 2.7191 Feasibility = 0.0000
>
> **GDMA ($\rho=10$)**
>
> iteration = 0001: Obj Loss = 2.4803 Feasibility = 1.0565
>
> iteration = 1000: Obj Loss = 3.1297 Feasibility = 0.0000
>
> iteration = 2000: Obj Loss = 3.4119 Feasibility = 0.0000
>
> iteration = 3000: Obj Loss = 3.5966 Feasibility = 0.0000
>
>
> We observe that our proposed approach reaches the lowest value for the first task loss among the approaches that satisfy the constraint within an accuracy $\epsilon=10^{-3}$. Note that while GDMA with $\rho=1$ yields a lower objective loss, it fails to meet the feasibility requirement within the desired accuracy threshold. The full details and corresponding plots for the experiment will be added to the revised manuscript.
>
> [Ref3] Jin, Chi, Praneeth Netrapalli, and Michael Jordan. "What is local optimality in nonconvex-nonconcave minimax optimization?." International conference on machine learning. PMLR, 2020.
>
> [Ref4] Nouiehed, Maher, et al. "Solving a class of non-convex min-max games using iterative first order methods." Advances in Neural Information Processing Systems 32 (2019).
>
>  ---
>
>  **4. Would Assumption 2.3 not hold for NNs in general?**
>
> **R4.**
> We thank the reviewer for this insightful question. While Assumption 2.3 holds in many practical NN architectures involving smooth components as noted in Remark 2.1, it does not extend to general NN architectures with nonsmooth components such as ReLU activation function, and one needs to use the smoothed variant, e.g., softplus, to ensure Assumption 2.3 holds.
>
>
>  ---
>  **5. Do the authors believe simplex constraints are representative enough for $\psi$ in the numerical experiments?**
>
> **R5.** We thank the reviewer for raising this question. As a simplex set describes a discrete probability distribution domain, it is a natural structure arising in distributionally robust models for capturing uncertainty when probabilities are unknown. Note that the regularization term in the robust MTL model considered in our numerical experiment finds the worst-case distributions in the neighborhood of the uniform distribution. This formulation is commonly used in DRO literature [Ref 5] and numerical examples for min-max optimization methods [Ref 6,Ref 7].
> Furthermore, the simplex set satisfies convexity and compactness, which help make the resulting optimization problem computationally manageable.
>
> [Ref 5] Namkoong, Hongseok, and John C. Duchi. "Stochastic gradient methods for distributionally robust optimization with f-divergences." Advances in neural information processing systems 29 (2016).
>
> [Ref 6] Zhang, Xuan, Necdet Serhat Aybat, and Mert Gurbuzbalaban. "Sapd+: An accelerated stochastic method for nonconvex-concave minimax problems." Advances in Neural Information Processing Systems 35 (2022): 21668-21681.
>
> [Ref 7] Mehta, Ronak, Jelena Diakonikolas, and Zaid Harchaoui. "DRAGO: Primal-Dual Coupled Variance Reduction for Faster Distributionally Robust Optimization." Advances in Neural Information Processing Systems 37 (2024): 134770-134825.
>
>  ---
>
> We hope our responses address your concerns and would be happy to clarify anything further if needed.

---

> > ### Comment · Reviewer_hjym · 2025-08-03
> >
> > Thank you for a thorough rebuttal and for answering each of my comments.
> >
> > One question. I do not think we should discard some literature because a minimization problem is different than a min-max problem. Min-max problems are still minimization problems. Conversely, some interesting functions can be reformulated with max-terms (e.g., with conjugate tricks). So, for a moment, can the authors ignore the "max" term inside (just assume we minimize a weird function that is defined with some max), and still claim that we cannot use ANY techniques from (D)RO? To my understanding, not having such a benchmark algorithm is unusual. (Note that I still appreciate this paper).

---

> > > ### Author Response · Authors · 2025-08-04
> > >
> > > We thank the reviewer for the comment and question. We acknowledge that there are techniques and methods that overlap between the DRO and min-max optimization domains, and we will include a discussion of these methods in the revised version of the paper. That said, we would like to clarify that while DRO reformulation techniques are powerful for structured settings, particularly when the ambiguity set admits tractable reformulations [Ref 1-8], they are not applicable for a general min-max problems. We also recognize the reviewer’s point that, under certain assumptions and ignoring the semi-infinite constraint, a min-max problem can be viewed as a minimization problem and if there are some structure to exploit indeed one can leverage DRO techniques to reformulate the problem as a constrained minimization problem. However, our experimental setup involves a constraint with infinite cardinality, resulting in a reformulated problem that remains a nonconvex semi-infinite program (SIP). To the best of our knowledge, there is no existing efficient algorithm with convergence guarantees for such problems, as current methods rely on stringent assumptions and often require solving non-trivial subproblems. Nevertheless, thanks to the reviewer's question, we added an additional baseline based on an adaptive discretization approach [Ref 9], where at each iteration, a set of sample points is selected to approximate the semi-infinite problem with a finite set of constraints (similar to the scenario approach in DOR [Ref 2,3]). The resulting constrained minimization subproblems are then solved using COOPER [Ref 10], an off-the-shelf optimization solver. However, this method failed to demonstrate effective convergence or competitive performance. Specifically, considering Multi-MNIST dataset our method’s output after 3000 iterations is
> > >
> > > Obj Loss = 1.9839,  Infeasibility = 0.0041
> > >
> > > While the output of the adaptive discretization method is
> > >
> > > Obj Loss = 3.8920,  Infeasibility = 2.3899.
> > >
> > > This underscoring the necessity of algorithmic solutions with theoretical guarantees, such as the one we propose. We will include the details of these numerical experiments and corresponding plots in the revised manuscript.
> > >
> > >
> > > [Ref 1] Kuhn, Daniel, Soroosh Shafiee, and Wolfram Wiesemann. "Distributionally robust optimization." Acta Numerica 34 (2025): 579-804.
> > >
> > > [Ref 2] Campi, Marco C., and Simone Garatti. "The exact feasibility of randomized solutions of uncertain convex programs." SIAM Journal on Optimization 19.3 (2008): 1211-1230.
> > >
> > > [Ref 3] Campi, Marco C., and Simone Garatti. "A sampling-and-discarding approach to chance-constrained optimization: feasibility and optimality." Journal of optimization theory and applications 148.2 (2011): 257-280.
> > >
> > > [Ref 4] De Farias, Daniela Pucci, and Benjamin Van Roy. "On constraint sampling in the linear programming approach to approximate dynamic programming." Mathematics of operations research 29.3 (2004): 462-478.
> > >
> > > [Ref 5] Mutapcic, Almir, and Stephen Boyd. "Cutting-set methods for robust convex optimization with pessimizing oracles." Optimization Methods & Software 24.3 (2009): 381-406.
> > >
> > > [Ref 6] Ben-Tal, Aharon, et al. "Oracle-based robust optimization via online learning." Operations Research 63.3 (2015): 628-638.
> > >
> > > [Ref 7] Tu, Kai, Zhi Chen, and Man-Chung Yue. "A max-min-max algorithm for large-scale robust optimization." arXiv preprint arXiv:2404.05377 (2024).
> > >
> > > [Ref 8] Postek, Krzysztof, and Shimrit Shtern. "First-order algorithms for robust optimization problems via convex-concave saddle-point Lagrangian reformulation." INFORMS Journal on Computing 37.3 (2025): 557-581.
> > >
> > > [Ref 9] Seidel, Tobias, and Karl-Heinz Küfer. "An adaptive discretization method solving semi-infinite optimization problems with quadratic rate of convergence." Optimization 71.8 (2022): 2211-2239.
> > >
> > > [Ref 10] Gallego-Posada, Jose, et al. "Cooper: A Library for Constrained Optimization in Deep Learning." arXiv preprint arXiv:2504.01212 (2025).

---

> > > > ### Comment · Reviewer_hjym · 2025-08-04
> > > >
> > > > Thank you for a thorough reply. The references are closer to what I had in mind. Please specify the distinction clearly in the camera-ready version. With that being said, I am increasing my score to 5. Congratulations on a great paper and good luck.

---

> > > > > ### Author Response · Authors · 2025-08-04
> > > > >
> > > > > We truly appreciate your support and will make sure to clearly highlight the distinction in the camera-ready version.

---

### Note · Authors · 2025-08-14

We thank all reviewers for their thoughtful and constructive feedback, which has improved the manuscript. During the rebuttal, we clarified our positioning within the semi-infinite programming literature, expanded the related work to include both classical and robust optimization references, and emphasized the distinctions between our nonconvex semi-infinite min-max setting and existing DRO formulations. We also addressed comments regarding applications, assumptions, and notation, and provided clearer explanations of theoretical conditions and algorithm parameters.

Our main contribution is a novel fully first-order algorithm with explicit global non-asymptotic convergence guarantees for nonconvex semi-infinite constrained min-max problems, offering iteration complexity bounds in terms of stationarity, feasibility, and complementarity slackness. These results, to the best of our knowledge, appear for the first time in the literature. We further enriched the numerical section with strong baselines, showing that our method consistently outperforms alternative formulations and a discretization method under the prescribed feasibility tolerance.

Finally, regarding the question about the relevance of the problem setting by reviewer E7VL, we note that semi-infinite programming with a min-max objective naturally arises in diverse applications such as robust learning, robotics, and reinforcement learning with safety constraints. Currently, applied researchers often simplify or reformulate such problems or resort to heuristic, problem-specific algorithms, due to the lack of efficient methods with provable guarantees. Our work addresses this gap and initiates a new line of research into first-order methods for nonconvex semi-infinite min-max problems, with significant potential to broaden their application in various domains.

---

### Decision · Program_Chairs · 2025-09-17

**Decision:**

Accept (poster)

**Comment:**

The paper studies a class of "semi-infinite nonconvex min-max optimization" problems, where the objective is nonconvex in the primal variables, strongly concave or satisfies the PL condition in the dual variables, and on the primal side is subject to a semi-infinite family of constraints that are possibly nonconvex but satisfy certain regularity conditions. This family of problems is motivated by applications in adversarial robustness and safety-constrained learning.

The paper introduces a first-order method that is guaranteed to converge to eps-approximate solutions (in terms of stationarity, complementary slackness, and feasibility) in poly(1/eps) iterations. While there is a broad literature on semi-infinite programming, this is the first result of its kind (first-order algorithm, nonasymptotic convergence guarantees). Despite initial concerns, there was broad support for the paper's acceptance, especially since there is potential for leading to further developments in this area. The concerns regarding practical impact of the results were downweighted, as the paper makes solid theoretical contributions, as pointed out in the reviews.